

# Photochemical aging of aerosols contributes significantly to the production of atmospheric formic acid

Yifan Jiang[1], Men Xia[2,3], Zhe Wang[4], Penggang Zheng[4], Yi Chen[4], and Tao Wang[1]

[1]Department of Civil and Environmental Engineering, The Hong Kong Polytechnic University, Hong Kong SAR 999077, China

[2]Institute for Atmospheric and Earth System Research/Physics, Faculty of Science, University of Helsinki, Helsinki 00014, Finland

[3]Aerosol and Haze Laboratory, Advanced Innovation Center for Soft Matter Science and Engineering, Beijing University of Chemical Technology, 100029, Beijing, China

[4]Division of Environment and Sustainability, Hong Kong University of Science and Technology, Hong Kong SAR 999077, China

*Correspondence to*: Tao Wang (tao.wang@polyu.edu.hk)

## Abstract

Formic acid (HCOOH) is one of the most abundant organic acids in the atmosphere and affects atmospheric acidity and aqueous chemistry. However, the formation mechanisms of HCOOH remain poorly understood, and current air-quality models largely underestimate observed atmospheric concentrations of HCOOH. In particular, HCOOH production from condensed-phase or heterogeneous reactions is not considered in current models. In a recent field study, we measured atmospheric HCOOH concentrations at a coastal site in South China. The average concentrations of HCOOH were $191.1 \pm 167.2$ ppt in marine air masses and $996.3 \pm 432.9$ ppt in coastal air masses. A strong linear correlation between HCOOH concentrations and the surface area densities of submicron particulate matter was observed in coastal air masses. Post-campaign laboratory experiments confirmed that the photochemical aging of ambient aerosols promoted by heterogeneous reactions with ozone produced a high concentration of HCOOH at a rate of 0.185 ppb h$^{-1}$ under typical ambient conditions at noon time. HCOOH production was strongly affected by nitrate photolysis, as this efficiently produces OH radicals that oxidise organics to form HCOOH. We incorporated this particle-phase source into a photochemical model and found that it explained 81% of the peak concentration of ambient HCOOH and reproduced the diurnal variation in HCOOH concentrations. These findings demonstrate that the photochemical aging of aerosols is an important source of HCOOH that must be included in atmospheric chemistry-transport models.



## 1. Introduction

Organic acids are ubiquitous in the troposphere and constitute a significant fraction of the total organics in both the gas and particle phases (Chebbi & Carlier, 1996). They also participate in the aqueous-phase chemistry of clouds, contribute to secondary organic aerosol (SOA) formation through reactions within the condensed phase (Carlton et al., 2007; Ervens et al., 2004; Lim et al., 2010), and are proposed to enhance the formation of new particles in the atmosphere (Zhang et al., 2004). Formic acid (HCOOH) is among the most abundant organic acids in the atmosphere (Khare et al., 1999) and accounts for over 60% of the free acidity in precipitation in remote areas and more than 30% of that in polluted areas (Andreae et al., 1988; Keene et al., 1983; Keene & Galloway, 1988; Khare et al., 1999; Stavrakou et al., 2012). This contribution is increasingly important due to the decline in the concentrations of anthropogenic nitrogen oxides ($NO_x$) and sulfur dioxide. HCOOH serves as a significant sink of in-cloud hydroxyl radicals (·OH) and stabilised Criegee intermediates (SCIs) (Jacob, 1986), and thus influences aqueous-phase chemistry by affecting pH-dependent reaction rates, oxidant concentrations, and solubilities (Vet et al., 2014). HCOOH also plays a role in the formation of cloud condensation nuclei (Yu, 2000), due to its comparatively higher hygroscopicity at low critical supersaturations when incorporated into aerosols (Novakov & Penner, 1993). This, in turn, affects total indirect radiative forcing. Additionally, HCOOH may be involved in halogen chemistry through its heterogeneous reaction with solid sodium chloride in sea-salt aerosols (Xia et al., 2018).

Considering the abovementioned roles of HCOOH in atmospheric chemistry, it is essential to understand its sources and sinks. However, the budget of HCOOH is currently poorly quantified, with state-of-the-art chemistry-transport models significantly underestimating field-observed concentrations of HCOOH (Baboukas et al., 2000; Bannan et al., 2017; Chaliyakunnel et al., 2016; Le Breton et al., 2012; Millet et al., 2015; Yuan et al., 2015). HCOOH is primarily removed from the atmosphere through wet and dry deposition, with a minor sink of being photo-oxidation by ·OH (Atkinson et al., 2006). The main sources of HCOOH include direct emissions from terrestrial vegetation (Andreae et al., 1988), biomass and biofuel burning (Akagi et al., 2011; Goode et al., 2000; Yokelson et al., 2009), fossil-fuel combustion (Kawamura et al., 2000; Zervas et al., 2001b, 2001a) and soil emissions (Sanhueza & Andreae, 1991). Moreover, secondary formation from the oxidation of volatile organic compounds (VOCs) is considered the major source of HCOOH at the global scale (Paulot et al., 2011). Despite the inclusion in models of various gas-phase mechanisms of HCOOH formation, such as ozonolysis of terminal alkenes (Neeb et al., 1997), alkyne oxidation (Bohn et al., 1996), OH-initiated isoprene oxidation (Paulot et al., 2009), monoterpene oxidation (Larsen et al., 2001), keto-enol tautomerisation (Andrews et al., 2012; Shaw et al., 2018) and ·OH oxidation of methyldioxy radicals (CH3O2·) (Bossolasco et al., 2014), HCOOH



concentrations remain significantly underestimated (Millet et al., 2015; Yuan et al., 2015),
indicating that a substantial missing source of HCOOH remains unidentified.
Current models do not consider HCOOH production from heterogeneous or condensed-
phase reactions, but these could be an important source of HCOOH. Aqueous reactions of
formaldehyde (HCHO) (Chameides & Davis, 1983; Jacob, 1986), glyoxal (Carlton et al., 2007),
and other species with ·OH (aq) can produce HCOOH, particularly in moderately acidic
environments (Jacob, 1986). A multiphase cloud-processing pathway involving methanediol
oxidation was proposed that reconciles model predictions with measured concentrations of
HCOOH (Franco et al., 2021). Moreover, Gao et al. (2022) recently proposed a new
bidirectional deposition-emission process, whereby HCOOH deposits rapidly in night-time
dew and is re-emitted from the dew as it evaporates on the following day. They found that this
process explained most of the concentrations of HCOOH that they observed. Laboratory
chamber studies have demonstrated that the photochemical aging of organic aerosols can also
produce HCOOH (Henry & Donahue, 2012; Malecha & Nizkorodov, 2016; Mang et al., 2008;
Pan et al., 2009; Walser et al., 2007; Zhang et al., 2021), but the importance of this process as
a source of atmospheric HCOOH has not been quantified, and this source is not considered in
current models.
The photochemical aging of aerosols occurs through the reactive uptake of oxidants onto
particle surfaces, altering their chemical compositions and physical properties (George et al.,
2015). In the condensed organic phase, this aging process can produce volatile compounds,
such as HCOOH, through the photodegradation of SOA (Henry & Donahue, 2012; Malecha &
Nizkorodov, 2016). Furthermore, the photolysis of particulate nitrate ($NO_3^-$) produces oxidants
such as ·OH, nitrogen dioxide ($NO_2$), and nitrite ions/nitrous acid (HONO), which efficiently
oxidise glyoxal to HCOOH (Zhang et al., 2021). Paulot et al. (2011) observed a marked positive
correlation between HCOOH concentrations and submicron organic aerosol masses in three
coastal, urban, and polar regions, and suggested that aerosol aging produces HCOOH. However,
another field and model study estimated that this aging process makes only a minor (<5%)
contribution to concentrations of HCOOH, although large uncertainties in the result were noted
(Yuan et al., 2015). Overall, the aforementioned results show that there is a need for the
determination of improved constraints on HCOOH production from the photochemical aging
of aerosols, as this will enable assessment of the significance of this process as a source of
HCOOH in comparison with other sources.
In this study, we measured HCOOH concentrations at near-ground level at a coastal site in
Hong Kong, China, for 2 months during autumn 2021. We examined the characteristics of
HCOOH concentrations and their correlation with related species' concentrations or other
parameters. We showed that the use of current gas-phase mechanisms in a photochemical box
model underpredicted the observed concentrations of HCOOH at our site. We then conducted
a chamber study to measure the rate of HCOOH production during the aging of ambient



aerosols and extrapolated the results to the real atmosphere. We incorporated this HCOOH-
formation mechanism into a model using a parameterisation involving fine particulate matter
concentration, surface area density, light intensity, and ozone ($O_3$) concentration, and then
performed simulations to evaluate the contribution made by the photochemical aging of
aerosols to HCOOH production. Furthermore, we showed that $NO_3^-$ photolysis acted as a
crucial source of ·OH during the aging process. Our results enhance the understanding of
HCOOH sources and model simulations of ambient HCOOH concentrations.

## 2. Methods

### 2.1. Field observations

Ambient measurements of the atmospheric concentrations of HCOOH and related
species/parameters were conducted from 13 August to 31 October 2021 at the Hong Kong
Environmental Protection Department's Cape D'Aguilar Super Site (CDSS; 22.21°N,
114.25°E), which is situated a few hundred meters away from the nearest coastline of the South
China Sea (Fig. S1). During the study period, this coastal site was initially predominantly
influenced by marine air masses and subsequently by coastal air masses. The site was also
affected by biogenic emissions (from deciduous and evergreen trees) and ship emissions (from
ships in nearby waters). There were no other significant anthropogenic sources nearby.
The species quantified were HCOOH, HONO, trace gases (nitric oxide, $NO_2$, carbon
monoxide (CO), sulfur dioxide ($SO_2$), and $O_3$), volatile organic compounds (VOCs),
oxygenated VOCs, aerosol mass concentrations (1-µm particulate matter ($PM_1$), $PM_{2.5}$, and
$PM_{10}$ concentrations), aerosol size distributions, aerosol ionic compositions, $NO_2$ photolysis
frequency ($jNO_2$), and meteorological parameters (temperature (T), relative humidity (RH),
wind direction, and wind speed). A detailed description of the measurement of HCOOH
concentration is provided below, and information on other measurements is summarised in
Table S1.
The concentration of HCOOH was measured at 172.91 atomic mass unit (amu) using an
iodide-adduct time-of-flight chemical ionisation mass spectrometer ($I^-$-ToF-CIMS, Aerodyne
Research), as we used in our previous study at the same site (M. Xia et al., 2022). A
comprehensive description of the use of the $I^-$-ToF-CIMS can be found in previous studies
(Aljawhary et al., 2013; Lee et al., 2014). Briefly, humidified iodomethane-containing $N_2$ air
was passed through an inline ioniser (containing polonium-210) to generate iodide ions ($I^-$) and
iodide–water ions, which served as the reagent ions, and the HCOOH produced was detected
as $IHCOOH^-$. The background concentration of HCOOH was determined every 2 days by
injecting zero air and was found to be 60.9 ppt. HCOOH calibration was performed three times
on-site and once in the laboratory immediately following the field campaign using diluted gas
standards generated by a permeation tube (KIN-TEK) with a permeation rate of 90.87 ng min$^-$



[1]. HCOOH sensitivity varies with RH as water competes with HCOOH for $I^-$ (Lee et al., 2014).
Thus, the HCOOH sensitivity was measured at various RHs, as shown in Fig. S2. The
sensitivity remained stable at a given RH, with a variation of less than 5% throughout the
campaign.
The ToF-CIMS was housed in an air-conditioned shelter at an indoor air T maintained at
25–28 °C. The shelter was located approximately 15 m away from the CDSS station. The
sampling tube was a 0.5-m long perfluoroalkoxy-Teflon tube (1/2 in. outer diameter), the inlet
of which was situated on the sidewall of the shelter, 1.5 m above the ground. To achieve laminar
flow in the sampling tube, a flow rate of 25 Lpm was adopted, with a residence time of 0.1 s.
The ToF-CIMS drew ~2 Lpm sample air, and the remaining airflow was discarded. The
sampling tube was replaced with a new tube every 2 days to reduce inlet artifacts. We
investigated possible inlet artifacts by injecting known concentrations of HCOOH into a used
sampling inlet and found that the artifacts had a negligible effect on the measured HCOOH
concentration (difference < 3%). The remaining instruments were housed in the CDSS station,
with their sampling inlets located ~1.5 m above the roof.
2.2. Hybrid Single-Particle Lagrangian Integrated Trajectory (HYSPLIT) and Extended
Aerosol Inorganic Model IV (E-AIM IV) models
Hourly 24-h backward trajectories were obtained using the HYSPLIT model
(https://www.ready.noaa.gov/hypub-bin/trajasrc.pl). The input parameters were Global Data
Assimilation System 1° for the meteorology data; 22.21°N and 114.25°E for the location; and
100 m for the endpoint height, which is slightly higher than the site's altitude. Air masses were
classified as marine or coastal, based on their source regions. A unique period was identified
as a haze period, during which there was a transition from marine air masses to coastal air
masses under stagnant conditions (wind speed < 3 m s$^{-1}$). A detailed classification can be found
in Fig. S3.
The aerosol water content (AWC) and pH were predicted by the E-AIM IV online in batch
mode (http://www.aim.env.uea.ac.uk/aim/model4/model4d.php). The thermodynamic model
was constrained by hourly field-measured molar concentrations of $NH_4^+$, sodium ions ($Na^+$),
sulfate ions ($SO_4^{2-}$), $NO_3^-$, and chloride ions ($Cl^-$) in $PM_{2.5}$; gas-phase ammonia concentrations;
ambient T; and RH. Initial concentrations of protons ($H^+$) and hydroxide ions were estimated
based on the ion balance of the major water-soluble ions. The input of $Na^+$ into the model
($Na^+_{(eq)}$) was calculated as a sum of the equivalent concentrations of $Na^+$, potassium ions,
magnesium ions, and calcium ions (Eq. (1)). The model also considered water dissociation and
allowed all possible solids to form in the system. Parameters p, q, r, and s (options in batch
mode) were set to 3, meaning that the input portions of ammonium ($NH_4^+$), $NO_3^-$, $SO_4^{2-}$, and
$Cl^-$ always remained in the condensed phase and did not produce corresponding gas-phase
species, such as nitric acid ($HNO_3$) and hydrochloric acid. The aqueous-phase $NO_3^-$
concentration was calculated by dividing the AWC by the moles of aqueous $NO_3^-$. Aerosol pH



was calculated as the negative logarithm of the concentration of $H^+$.
$$[Na^+]_{eq} = [Na^+] + [K^+] + 2[Mg^{2+}] + 2[Ca^{2+}] \qquad (1)$$
2.3. Chemical box model
The Framework for 0-D Atmospheric Modeling (F0AM version 4.2.1) (Wolfe et al., 2016)
coupled with Master Chemical Mechanism (MCM v3.3.1, http://mcm.york.ac.uk) (Jenkin et
al., 2015) was employed to investigate the formation of HCOOH at the field site. We followed
Yuan et al. (2015) by enhancing the simulation of the secondary formation of HCOOH by
modifying the HCOOH yields obtained by the ozonolysis of alkenes and other unsaturated
species, and by adding chemical processes (vinyl alcohol oxidation; photo-tautomerisation of
vinyl alcohol; and the reactions of HCHO + $HO_2$ and $CH_3O_2$· + ·OH) to the MCM. A single
typical episode day, 28 September 2021, was selected as an example to run the model, as the
critical model-input data were available on this day. The measured concentrations of $O_3$, $NO_x$,
CO, $SO_2$, HONO, VOCs, and OVOCs were averaged or interpolated to 5-min resolution and
constrained in the model. We also run the model using hourly averaged data of another typical
day in coastal air masses, 28 October 2021, and diurnal variations during the whole campaign.
The methane concentration was assumed to remain constant (2000 ppb; Peng et al., 2022), due
to a lack of measurement data. The dilution process was treated as a first-order loss with a
dilution loss rate coefficient of 1/86,400 $s^{-1}$, consistent with previous studies (Li et al., 2014;
Yuan et al., 2015). The dry deposition rate was determined based on the deposition velocity
and the boundary layer height (BLH). For HCOOH, a deposition velocity of 1 cm $s^{-1}$ was
employed (Müller et al., 2018). A sensitivity analysis of the deposition velocity was also
conducted. The diurnal profile of BLH was acquired from a previous study at another coastal
site in Hong Kong (Su et al., 2017). Wet deposition was not considered as there was no rainfall
on 28 September 2021. The model was executed for three replicates to stabilise the intermediate
species it generated, and the results from the final run were used for further analysis. Primary
emissions and transportation from other regions were not considered in the box model; thus,
the production rate of HCOOH was utilised to evaluate the significance of various chemical
pathways.
2.4. Laboratory experiments
We illuminated ambient aerosols collected on filters or aqueous solutions in a dynamic
chamber to mimic HCOOH formation in the atmosphere. The overall experimental setup is
illustrated in Fig. S4. The dynamic chamber has a dimension of 25-cm length × 15-cm width ×
4-cm height with the top side sealed by a transparent Teflon film. Each aerosol filter or solution
sample was placed in a quartz Petri dish (inner diameter: 35 mm, inner height: 7 mm) at the
chamber's centre. Aerosol filter sampling details can be found in Text S1. A high-pressure
xenon (Xe) lamp was used to simulate sunlight and its spectral irradiance is displayed in Fig.
6d. Compared with standard air mass 1.5 solar irradiation (AM 1.5) corresponding to a solar
zenith angle of 48.2°, the Xe lamp exhibited a smaller flux at 300–326 nm but a larger flux at



326–420 nm. An air stream from a zero-air generator (Environics, model 7000) served as the carrier gas that delivered reaction products to the chamber's outflow. The weather conditions that prevailed during the field campaign were mimicked by maintaining the reactor's T at approximately 28 ℃ and adjusting the RH in the chamber to 70% by passing the carrier gas through a water bubbler. Prior to the introduction of a sample into the chamber, the background HCOOH concentration was monitored for 10 min with the light on and zero air added. After 1 h of irradiation, 100 ppb of $O_3$ was introduced via a dynamic calibrator (Environics® Series 6100) and monitored using an $O_3$ analyser (Thermo Scientific Model 49i). An AM 1.5 filter (which removes light below 360 nm) and a 300–800 nm filter (which allows the passage of 300-800 nm light) were applied to the Xe lamp to investigate the effect of the irradiation wavelength on the formation of HCOOH.

A sample solution was prepared by mixing formaldehyde (HCHO, Sigma-Aldrich, 37 wt% in water) and sodium nitrate (NaNO₃, Honeywell, 99.5% purity). The resulting solution contained 0.15 wt% HCHO and 0.2 M NaNO₃, and was adjusted to pH 2.7 by the addition of sulfuric acid ($H_2SO_4$, Sigma-Aldrich, 98% purity), as this was the E-AIM model's prediction of the average aerosol acidity during the entire campaign. We assumed that ·OH produced by $NO_3^-$ photolysis was the rate-limiting species and HCHO was taken as an example of one of the possible precursors of HCOOH.

The average rate of production (ppb·s⁻¹) of HCOOH ($P_{HCOOH}$) during the 1-h irradiation was calculated by the following equation (Eq. (2)), derived from (Peng et al., 2022):

$$P_{HCOOH} = \int_0^{60} (C_{HCOOH} - C_{HCOOH-bkg}) \, dt \times \frac{Q}{V} / 60 \qquad (2)$$

where Q is the carrier gas flow rate (4 L min⁻¹); V is the reactor chamber volume (1.875 L); and $C_{HCOOH}$ and $C_{HCOOH-bkg}$ (ppb) are the concentrations of HCOOH in the chamber after and before adding the sample, respectively. The photolytic loss of HCOOH was ignored, as the cross-section of HCOOH was beyond the spectral range of the Xe lamp (Burkholder et al., 2020).

We attempted to extrapolate the laboratory results to account for the field-observed concentrations of HCOOH. As photochemical aging occurs on aerosol surfaces and a strong correlation between the surface area (Sa) and the concentration of HCOOH observed in the field, the extrapolation was conducted based on Sa. The Sa in the chamber was calculated as the Sa of the filter divided by the chamber's volume, assuming that only the first layer of the aerosols was illuminated. Although this might have resulted in an underestimation of the Sa density in the chamber, this assumption was reasonable, because particles in the lower layers would receive less light than those in the uppermost layer due to the light-screening effect of the first layer (Ye et al., 2017). The aging process was also influenced by light intensity, and as we discovered that the major oxidant was generated by the photolysis of particulate $NO_3^-$ (See **Results**, section 3), the light intensity was normalised based on the photolytic frequency of





aqueous $NO_3^-$ ($J_{NO_3^-(aq)}$) due to the absence of an absorption coefficient for particulate $NO_3^-$.
Although there is a redshift of the particulate $NO_3^-$ absorption wavelength compared with the
aqueous-phase $NO_3^-$ absorption wavelength, our results should be reliable because we used
$J_{NO_3^-(aq)}$ as a reference for normalisation rather than for calculating an accurate $J_{NO_3^-(aq)}$ (Du
& Zhu, 2011; Zhu et al., 2008). The $J_{NO_3^-(aq)}$ under the Xe lamp was $8.85 \times 10^{-6}$ s$^{-1}$ and the
daytime average $J_{NO_3^-(aq)}$ in the ambient air at our site was $1.12 \times 10^{-5}$ s$^{-1}$ (Text S3). Both
$J_{NO_3^-(aq)}$ values were calculated assuming a quantum yield equal to 1. The normalised HCOOH
production rate in the ambient air ($P_{HCOOH-nml}$) was calculated using the following equation (Eq
264 (3)):

$P_{HCOOH-nml} = P_{HCOOH} \times \frac{Sa_{amb}}{Sa_{cha}} \times 1.266$ (3)
where $Sa_{amb}$ represents the field-measured Sa density; $Sa_{cha}$ denotes the Sa density calculated
for the chamber; and 1.266 is the ratio of the ambient $J_{NO_3^-(aq)}$ to the chamber $J_{NO_3^-(aq)}$. For the
aging process involving $O_3$, the photolytic rate constant of $O_3$ generating $O^1D$ ($J_{O_3 \to O^1D}$) in the
chamber ($1.31 \times 10^{-5}$ s$^{-1}$) was also normalised to the average daytime $J_{O_3 \to O^1D}$ ($1.84 \times 10^{-5}$ s$^{-1}$) under ambient conditions (Text S3). For the results of the aqueous solution, the
concentrations of HCHO and $NO_3^-$ were also normalised.

## 3. Results and Discussion

### 3.1. Field measurements of HCOOH concentrations

The field site was exposed to two distinct types of air masses; initially, it was largely
exposed to marine air masses, and later to coastal air masses. Marine air masses (T = 29.4 ±
2.0 °C, RH = 85.8 ± 7.0 %) were warmer and more humid than coastal air masses (T = 25.7 ±
2.3 °C, RH = 77.0 ± 6.0 %), and exhibited low concentrations of $O_3$ (15.0 ± 8.9 ppb) and high
concentrations of $NO_x$ (6.2 ± 4.5 ppb). Conversely, coastal air masses were characterised by
high concentrations of $O_3$ (53.6 ± 14.2 ppb) and low concentrations of $NO_x$ (1.9 ± 1.6 ppb).
The high concentrations of $NO_x$ in the marine air masses are attributable to the emissions from
ocean-going container ships that passed the site approximately 8 km to the south. A haze event
occurred from 24 September to 2 October, due to a transition from marine to coastal air masses
under stagnant conditions. $O_3$ concentrations steadily increased during the first 5 days, peaked
on 29 September, and remained high until the end of the haze period (Fig. 1). Therefore, the
potential HCOOH formation mechanism was analysed separately for these three distinct
periods.
Ambient HCOOH concentrations significantly varied during the three periods. The average
HCOOH concentration in marine air masses was 191.1 ± 167.2 ppt; this was higher than those
over the remote ocean, due to local emission sources, but significantly lower than those in
urban environments (Table 1). In contrast, the ambient HCOOH concentrations in coastal air



masses were substantially higher, averaging 996.3 ± 432.9 ppt, comparable with other measurements at rural or urban background sites. During the haze period, the concentrations of HCOOH displayed a pattern similar to the concentrations of $O_3$, with the daytime peak concentration increasing from 673.5 to 2789.9 ppt. A pronounced diurnal variation in the concentration of HCOOH was observed throughout the entire campaign, as illustrated in Fig. 2, consistent with other studies (Millet et al., 2015; Yuan et al., 2015). HCOOH concentrations rapidly increased after sunrise, peaking at approximately 1 pm (local time), and then quickly decreasing in the late afternoon, due to the weaker sunlight and lower BLH than earlier in the day.

HCOOH is widely recognised as a secondary photochemical product. Table 2 presents the Pearson correlation coefficients ® between the concentration of HCOOH and those of other air pollutants or other meteorological parameters during the three distinct periods. The concentration of $HNO_3$ was strongly correlated with the concentration of HCOOH throughout the entire field campaign, consistent with other studies (Bannan et al., 2017; Millet et al., 2015). This finding suggests that HCOOH is predominantly generated through secondary photochemical mechanisms at this site, as $HNO_3$ is a secondary photochemical product resulting from the reaction between ·OH and $NO_2$. The positive linear relationship between the concentrations of $O_3$ and HCOOH also implies the secondary source of HCOOH.

A previous laboratory study revealed that HCOOH can be produced by the photochemical aging of aerosols (Malecha & Nizkorodov, 2016), which may be an important process in ambient air. In the coastal air masses and haze period, there was a strong correlation between the concentrations of HCOOH and PM, particularly between the concentrations of HCOOH and $PM_1$. This was also observed by Paulot et al. (2011) and suggests that HCOOH may be produced from PM. The Sa of $PM_1$ was also highly correlated with the concentration of HCOOH in both coastal air masses and haze periods, indicating that HCOOH is mainly produced from reactions on aerosol surfaces. However, in the marine air masses, the concentration of HCOOH was not related to aerosols due to the low particle concentrations in such masses. To further explore the potential role of aerosol aging in HCOOH production, we plotted the correlation of HCOOH concentrations with $Sa \times O_3$, $Sa \times NO_3^-$, and $Sa \times O_3 \times NO_3^-$ for the coastal air masses (Fig. 3). We discovered that the correlation coefficient significantly increased when Sa was combined with the concentration of $O_3$ or $NO_3^-$ or with the concentrations of both species, compared with these three factors being considered separately. This finding suggests that the HCOOH observed in the coastal air masses was not predominantly derived from gas-phase $O_3$ oxidation of VOCs; rather, it was derived from heterogeneous or condensed-phase reactions on aerosol surfaces. The results during the haze period were similar. Therefore, photochemical aerosol aging may play a key role in HCOOH production as the aging process involves the reactive uptake of oxidants onto particle surfaces.





### 3.2. Box model simulation

A box model was utilised to evaluate the formation mechanisms of HCOOH using the measurement data from a typical haze day (28 September 2021). The peak HCOOH concentration on that day was approximately 2 ppb, and occurred at approximately 3:30 pm. The base model, incorporating only the default mechanism of MCM v3.3.1, significantly underestimated the HCOOH concentration: the highest simulated concentration was 0.256 ppb, representing only 14.5% of the observed value. We made modifications to the formation mechanisms following Yuan et al. (2015) and these resulted in the simulated peak daytime concentration increasing to 0.363 ppb, accounting for 20.1% of the observed value (Fig. 4). Therefore, an additional HCOOH formation mechanism is required to account for the difference between the measured and simulated values.

A comprehensive analysis of HCOOH sources and sinks was conducted for both the base and modified cases (Fig. 5a & 5b). The reaction of $CH_2OO$ Criegee intermediate biradicals with $H_2O$ was identified as the major source of HCOOH, accounting for over 90% of the current known sources for both cases. $CH_2OO$ is formed from seven excited biradicals that originate from the $O_3$ oxidation of various alkenes and unsaturated compounds (Saunders et al., 2003). Among these, $CH_2OOE$ is the largest contributor to the production of $CH_2OO$ (Fig. S5) and is generated by the ozonolysis of isoprene. The primary loss of HCOOH is via deposition, owing to its high solubility in water. To account for uncertainty in the deposition velocity ($V_d$) of HCOOH, we conducted a sensitivity test of HCOOH production to various $V_d$ values (Fig. 5d). The results revealed that the simulated HCOOH concentration was insensitive to $V_d$ when it was higher than 1.00 cm s$^{-1}$. The daytime peak concentration increased by 60% when $V_d$ decreased from 1.00 to 0.50 cm s$^{-1}$, but the model still largely underestimated the HCOOH concentration. The field-observed $V_d$ of HCOOH ranges from 0.43 cm s$^{-1}$ to 1.10 cm s$^{-1}$ (Müller et al., 2018), and thus given the high humidity at the study site, the observed $V_d$ of HCOOH of 0.5 cm s$^{-1}$ should have been close to the lower limit. The simulated net HCOOH production (sources – sinks) became positive at approximately 9 am, while the ambient concentration of HCOOH started increasing at 6 am, which is aligned with sunrise (Fig. 2). These results indicate that there are pathways for the photochemical generation of HCOOH that are distinct from $O_3$ oxidation and these may include the photochemical aging of aerosols. We also executed the model on 28 October 2021, another day that was exposed to coastal air masses, and obtained similar outcomes (Fig. S6).

### 3.3. Laboratory experiments

Figure 6a presents the results of a typical aerosol-filter irradiation experiment. Upon turning on the light, HCOOH was instantaneously produced, indicating a rapid transfer from the condensed-phase to the gas phase through photochemical reactions. Within 3 minutes, the HCOOH concentration reached 11.1 ppb, but the when the light was turned off, the HCOOH concentration quickly returned to nearly background concentrations. This suggests that



HCOOH was produced predominantly via photochemical reactions. The HCOOH
concentration exhibited a logarithmic decay after its first peak concentration and this decay
also occurred continued after its second peak concentration, which may be attributable to either
the evaporative loss of HCOOH or the photochemical loss of oxidants (Ye et al., 2017). When
the AM 1.5 filter was added, the HCOOH concentration decreased by approximately 48.1%
within 5 min, and after the filter was removed, the HCOOH concentration returned to the
logarithmic decay line. This suggests that there was only minor evaporation of HCOOH from
the condensed-phase due to the increased temperature of aerosol surfaces under light irradiation.
However, the addition of the 300–800 nm filter reduced the HCOOH concentration by only
13.2%, indicating that the photochemical production of HCOOH primarily occurs at
wavelengths lower than 360 nm. Given the agreement between the wavelength at which $NO_3^-$
absorbs light (290–350 nm) and the wavelength of HCOOH production (<360 nm), and the
high correlation between the ambient HCOOH concentration and the product of Sa density and
$NO_3^-$ concentration (as shown in Fig. 3), we infer that ·OH produced from $NO_3^-$ photolysis
were the major oxidants in the particle phase and thus drove HCOOH production. The
production of HCOOH was also found to be dependent on $O_3$, as the concentration of HCOOH
increased by 64.7% after the addition of 100 ppb of $O_3$.

383       We next extrapolated the production rate of HCOOH observed in the chamber to ambient
conditions, using the method described in Section 2.4, to assess the role played by the
photochemical aging of aerosols in HCOOH production. Table 3 summarises the HCOOH
concentrations and production rates observed in the chamber experiments, and the normalised
HCOOH production rates in ambient air under light and light + $O_3$ conditions, respectively.
The average $P_{HCOOH-nml}$ without the addition of $O_3$ was determined to be 0.106 ppb h$^{-1}$,
equivalent to 138.5% of the peak HCOOH production rate in the modified case. The addition
of 100 ppb of $O_3$ increased $P_{HCOOH-nml}$ by 0.079 ppb h$^{-1}$, indicating that the heterogenous
reaction between $O_3$ and aerosols made a non-negligible contribution to HCOOH production.
By comparing the net $P_{HCOOH-nml}$ via the photochemical aging pathway with and without $O_3$, it
was found that the incorporation of these two conditions into the model should improve the
model results by factors of 2.89 and 1.51, respectively. These results highlight the importance
of HCOOH production via the aging of aerosols, which we found generated more HCOOH
than gas-phase reactions at our observation site.

397       We established a relationship between $P_{HCOOH-nml}$ and three parameters: $PM_{2.5}$
concentration (cPM), which represents the reactant concentration; Sa, which represents the
available reaction area; and jNO$_2$, which represents the light intensity. After multiplying these
three factors, we discovered a strong linear correlation between $P_{HCOOH-nml}$ and cPM × Sa ×
jNO$_2$ (Fig. 6b). The intercept was set to zero, as there should be no HCOOH production when
cPM × Sa × jNO$_2$ is zero. Based on the correlation, we derived an equation (Eq. (4)) for
calculating $P_{HCOOH-nml}$. Additionally, we assumed that $P_{HCOOH-nml (O3)}$ increased linearly with $O_3$



concentration. Incorporating this equation into the F0AM model by treating the photochemical
production of HCOOH from particles as an emission resulted in significantly improved
predictions: they explained 81% of the peak concentration, as illustrated by the black line in
Figure 4. The production of HCOOH from particles was the largest source of HCOOH,
accounting for 76% of the total production (Fig. 5c). Moreover, the model also reproduced a
rapid increase in the concentration of HCOOH in the morning and a sharp decrease in the
concentration of HCOOH in the late afternoon (Fig. 4). The different trend at midnight is
attributable to the continuous deposition of HCOOH concomitant with no production in the
model at that time. We also evaluated the model's performance in simulating HCOOH
production for 28 October 2021. The particle-phase production narrowed the gap significantly
and constituted over 70% of the total production (Fig. S6), which is consistent with the results
for 28 September 2021.
In summary, the inclusion of HCOOH production from the photochemical aging of aerosols
significantly improved the performance of the model. Although there may be limitations to the
assumptions made in the parameterisation, particularly in environments containing with
different chemical compositions of particles, this new parameterisation provides a general form
constrained by four factors for calculating HCOOH production from the condensed phase.
$$P_{HCOOH-nml} = 0.0091x + 0.010_3 \times 0.0064x, \quad x = cPM \times Sa \times jNO_2 \qquad (4)$$
Photolysis of particulate $NO_3^-$ is an important source of ·OH (Mack & Bolton, 1999;
Zellner et al., 1990). To investigate the potential production of HCOOH from this source, an
irradiation experiment was conducted on a solution (Fig. 6c). The concentration of HCOOH
increased linearly with time and did not reach a stable state after 90 min of illumination. This
differs from the aerosol experiments and might have been due to the continuous evaporation of
water from the solution caused by the heating effect of the light source, which would have
concentrated the solution. To determine the appropriate time to calculate $P_{HCOOH}$, we also
plotted the time series of HONO concentrations. This showed that the HONO concentration
stabilised after 1 h of irradiation, suggesting that $NO_3^-$ photolysis also reached a steady state.
As ·OH produced from $NO_3^-$ photolysis were the only oxidants present in the system, the actual
HCOOH production rate at the initial HCHO concentration should have followed the same
trend as the HONO concentration. Therefore, we chose 1 h after turning on the light as the
appropriate time to quantify $P_{HCOOH}$, and found that at this time, $P_{HCOOH}$ in the chamber was
21.9 ppt s$^{-1}$. We also attempted to extrapolate the results to ambient air, similar to the aerosol
filter experiments. To do so, in addition to normalising Sa and light intensity, we needed to
normalise the HCHO and $NO_3^-$ concentrations. The average concentration of gas-phase HCHO
(HCHO$_{(g)}$) measured on 28 September was 2.35 µg cm$^{-3}$. By using a ratio of 0.03 between the
concentration of HCHO in the particle phase (HCHO$_{(p)}$) and the concentration of HCHO in the
gas phase (Toda et al., 2014), the concentration of HCHO$_{(p)}$ was calculated to be 0.07 µg cm$^{-3}$,
which is comparable to the concentrations that have been reported in previous studies (Klippel



& Warneck, 1980; Toda et al., 2014). Based on the aqueous volume of aerosol calculated by
the E-AIM model (0.02 µl m$^{-3}$), the HCHO mass concentration in the aqueous phase was found
to be 3.5 g L$^{-1}$. The NO$_3^-$ concentration on the aerosol surface was determined to be 0.98 mol
L$^{-1}$. Therefore, the P$_{HCOOH}$ in ambient air attributable to the aqueous oxidation of HCHO was
estimated to be 0.41 ppb h$^{-1}$, which is 285% higher than the P$_{HCOOH}$ attributable to the
photochemical aging of ambient particles. This greater-than-100% contribution could be
attributable to the simplicity of the solution system. In ambient air, other oxidisable species,
such as halides (Peng et al., 2022; M. Xia et al., 2022) could also react with the ·OH produced
from NO$_3^-$ photolysis, which competes with HCHO. In summary, NO$_3^-$ photolysis appears to
be a critical source of oxidants during the photochemical aging process of aerosols.

## 4. Conclusion and implications

This study provides the first estimate of high rates of HCOOH production from the
photochemical aging of real ambient particles and demonstrates the potential importance of
this pathway in the formation of HCOOH under ambient conditions at a coastal site in Hong
Kong. Incorporating aerosol photochemical aging significantly improved the performance of a
widely used chemical model, which underscores the significance of condensed-phase
photochemistry and the necessity of incorporating its mechanisms into atmospheric models.
The substantial production of HCOOH from condensed-phase photochemical reactions altered
both the composition and the volatility of SOA. Moreover, other low-molecular-weight organic
acids, such as acetic acid, may be produced via this mechanism, as observed in our irradiation
experiments. Improving the constraints on this photochemical aging of aerosols will not only
aid the understanding of the budget of these organic acids but also affect their SOA chemistry.
Our solution irradiation experiments demonstrated the importance of NO$_3^-$ photolysis in
HCOOH production via the production of ·OH. This suggests that NO$_3^-$ photolysis not only
influences the aerosol-based production of inorganic species (such as HONO (Ye et al., 2017)
and halogens (Peng et al., 2022; M. Xia et al., 2022), but also the aerosol-based production of
organics, particularly water-soluble organics. Thus, there is a need for future studies on the
roles of NO$_3^-$ photodissociation in aerosol aging processes, as such research would improve
our understanding of the aging mechanisms of the condensed phase.

## Data availability

The data that support the findings of this study are openly available in Zenodo at
https://doi.org/10.5281/zenodo.8059231. Other raw data are also available from the
corresponding author, upon reasonable request.



## Authors' contributions

T. W. arranged the field campaign and designed the laboratory irradiation experiment. Y. J. and M. X. conducted the field campaign and photochemical filter experiments. M. X. revised the model code. Y. J. conducted the photochemical solution experiments, analysed the data, ran the model, and wrote the draft manuscript. T. W. and M. X. revised the manuscript.

## Competing interests

One author (Tao Wang) is a member of the editorial board of Atmospheric Chemistry and Physics. The authors have no other competing interests to declare.

## Acknowledgements

We thank the Hong Kong Environmental Protection Department for allowing us to use the field study site and for providing data on VOCs, OVOCs, trace gases, PM mass concentrations, and ion compositions; the Hong Kong Observatory for providing the meteorological data; the Hong Kong Polytechnic University Research Facility in Chemical and Environmental Analysis for providing the ToF-CIMS; and Dr Zhao Jue for providing $PM_{2.5}$ filters, whose work is supported by projects (PolyU Project of Strategic Importance No. ZE2K and RGC-GRF No. 15203920). We are grateful to Steven Poon for his help with logistics.

## Funding

This work was supported by the Research Grants Council of the Hong Kong (Project No. T24/504/17).

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

A Large Underestimate of Formic Acid from Tropical Fires: Constraints from Space-Borne
Measurements. Environmental Science and Technology, 50(11), 5631–5640.
https://doi.org/10.1021/acs.est.5b06385
Chameides, W. L., & Davis, D. D. (1983). Aqueous-phase source of formic acid in clouds.
Nature 1983 304:5925, 304(5925), 427–429. https://doi.org/10.1038/304427a0
Chebbi, A., & Carlier, P. (1996). Carboxylic acids in the troposphere, occurrence, sources, and
sinks: A review. Atmospheric Environment, 30(24), 4233–4249. https://doi.org/10.1016/1352-



544    2310(96)00102-1

Chen, X., Millet, D. B., Neuman, J. A., Veres, P. R., Ray, E. A., Commane, R., Daube, B. C.,
McKain, K., Schwarz, J. P., Katich, J. M., Froyd, K. D., Schill, G. P., Kim, M. J., Crounse, J.
D., Allen, H. M., Apel, E. C., Hornbrook, R. S., Blake, D. R., Nault, B. A., … Dibb, J. E. (2021).
HCOOH in the Remote Atmosphere: Constraints from Atmospheric Tomography (ATom)
Airborne Observations. ACS Earth and Space Chemistry, 5(6), 1436–1454.
https://doi.org/10.1021/acsearthspacechem.1c00049
Du, J., & Zhu, L. (2011). Quantification of the absorption cross sections of surface-adsorbed
nitric acid in the 335-365 nm region by Brewster angle cavity ring-down spectroscopy.
Chemical Physics Letters, 511(4–6), 213–218. https://doi.org/10.1016/j.cplett.2011.06.062
Ervens, B., Feingold, G., Frost, G. J., & Kreidenweis, S. M. (2004). A modeling study of
aqueous production of dicarboxylic acids: 1. Chemical pathways and speciated organic mass
production. Journal of Geophysical Research: Atmospheres, 109(D15).
https://doi.org/10.1029/2003JD004387
Fulgham, S. R., Brophy, P., Link, M., Ortega, J., Pollack, I., & Farmer, D. K. (2019). Seasonal
Flux Measurements over a Colorado Pine Forest Demonstrate a Persistent Source of Organic
Acids. ACS Earth and Space Chemistry, 3(9), 2017–2032.
https://doi.org/10.1021/acsearthspacechem.9b00182
George, C., Ammann, M., D'Anna, B., Donaldson, D. J., & Nizkorodov, S. A. (2015).
Heterogeneous Photochemistry in the Atmosphere. Chemical Reviews, 115(10), 4218–4258.
https://doi.org/10.1021/CR500648Z
Goode, J. G., Yokelson, R. J., Ward, D. E., Susott, R. A., Babbitt, R. E., Davies, M. A., & Hao,
W. M. (2000). Measurements of excess O3, CO2, CO, CH4, C2H4, C2H2, HCN, NO, NH3,
HCOOH, CH3COOH, HCHO, and CH3OH in 1997 Alaskan biomass burning plumes by
airborne Fourier transform infrared spectroscopy (AFTIR). Journal of Geophysical Research:
Atmospheres, 105(D17), 22147–22166. https://doi.org/10.1029/2000JD900287
Henry, K. M., & Donahue, N. M. (2012). Photochemical aging of α-pinene secondary organic
aerosol: Effects of OH radical sources and photolysis. Journal of Physical Chemistry A,
116(24), 5932–5940. https://doi.org/10.1021/JP210288S
Jacob, D. J. (1986). Chemistry of OH in remote clouds and its role in the production of formic
acid and peroxymonosulfate. Journal of Geophysical Research, 91(D9), 9807.
https://doi.org/10.1029/JD091ID09P09807
Jenkin, M. E., Young, J. C., & Rickard, A. R. (2015). The MCM v3.3.1 degradation scheme for
isoprene. Atmospheric Chemistry and Physics, 15(20), 11433–11459.
https://doi.org/10.5194/ACP-15-11433-2015
Kawamura, K., Steinberg, S., & Kaplan, I. R. (2000). Homologous series of C1-C10
monocarboxylic acids and C1-C6 carbonyls in Los Angeles air and motor vehicle exhausts.
Atmospheric Environment, 34(24), 4175–4191. https://doi.org/10.1016/S1352-



582    2310(00)00212-0

Keene, W. C., & Galloway, J. N. (1988). The biogeochemical cycling of formic and acetic acids
through the troposphere: an overview of current understanding. Chemical and Physical
Meteorology, 40(5), 322–334. https://doi.org/10.3402/tellusb.v40i5.15994
Keene, W. C., Galloway, J. N., & Holden, J. D. (1983). Measurement of weak organic acidity
in precipitation from remote areas of the world. Journal of Geophysical Research, 88(C9),
5122–5130. https://doi.org/10.1029/JC088IC09P05122
Khare, P., Kumar, N., Kumari, K. M., & Srivastava, S. S. (1999). Atmospheric formic and
acetic acids: An overview. Reviews of Geophysics, 37(2), 227–248.
https://doi.org/10.1029/1998RG900005
Klippel, W., & Warneck, P. (1980). The formaldehyde content of the atmospheric aerosol.
Atmospheric Environment (1967), 14(7), 809–818. https://doi.org/10.1016/0004-
594    6981(80)90137-7

Larsen, B. R., di Bella, D., Glasius, M., Winterhalter, R., Jensen, N. R., & Hjorth, J. (2001).
Gas-Phase OH Oxidation of Monoterpenes: Gaseous and Particulate Products. Journal of
Atmospheric Chemistry 2001 38:3, 38(3), 231–276. https://doi.org/10.1023/A:1006487530903
le Breton, M., McGillen, M. R., Muller, J. B. A., Bacak, A., Shallcross, D. E., Xiao, P., Huey,
L. G., Tanner, D., Coe, H., & Percival, C. J. (2012). Airborne observations of formic acid using
a chemical ionization mass spectrometer. Atmospheric Measurement Techniques, 5(12), 3029–
3039. https://doi.org/10.5194/amt-5-3029-2012
Lee, B. H., Lopez-Hilfiker, F. D., Mohr, C., Kurtén, T., Worsnop, D. R., & Thornton, J. A.
(2014). An iodide-adduct high-resolution time-of-flight chemical-ionization mass spectrometer:
Application to atmospheric inorganic and organic compounds. Environmental Science and
Technology, 48(11), 6309–6317. https://doi.org/10.1021/es500362a
Li, X., Rohrer, F., Brauers, T., Hofzumahaus, A., Lu, K., Shao, M., Zhang, Y. H., & Wahner, A.
(2014). Modeling of HCHO and CHOCHO at a semi-rural site in southern China during the
PRIDE-PRD2006 campaign. Atmospheric Chemistry and Physics, 14(22), 12291–12305.
https://doi.org/10.5194/ACP-14-12291-2014
Lim, Y. B., Tan, Y., Perri, M. J., Seitzinger, S. P., & Turpin, B. J. (2010). Aqueous chemistry
and its role in secondary organic aerosol (SOA) formation. Atmospheric Chemistry and Physics,
10(21), 10521–10539. https://doi.org/10.5194/ACP-10-10521-2010
Mack, J., & Bolton, J. R. (1999). Photochemistry of nitrite and nitrate in aqueous solution: a
review. Journal of Photochemistry and Photobiology A: Chemistry, 128(1–3), 1–13.
https://doi.org/10.1016/S1010-6030(99)00155-0
Malecha, K. T., & Nizkorodov, S. A. (2016). Photodegradation of Secondary Organic Aerosol
Particles as a Source of Small, Oxygenated Volatile Organic Compounds.
https://doi.org/10.1021/acs.est.6b02313
Mang, S. A., Henricksen, D. K., Bateman, A. E., Andersen, M. P. S., Blake, D. R., &



Nizkorodov, S. A. (2008). Contribution of carbonyl photochemistry to aging of atmospheric
secondary organic aerosol. Journal of Physical Chemistry A, 112(36), 8337–8344.
https://doi.org/10.1021/JP804376C
Millet, D. B., Baasandorj, M., Farmer, D. K., Thornton, J. A., Baumann, K., Brophy, P.,
Chaliyakunnel, S., de Gouw, J. A., Graus, M., Hu, L., Koss, A., Lee, B. H., Lopez-Hilfiker, F.
D., Neuman, J. A., Paulot, F., Peischl, J., Pollack, I. B., Ryerson, T. B., Warneke, C., … Xu, J.
(2015). A large and ubiquitous source of atmospheric formic acid. Atmospheric Chemistry and
Physics, 15(11), 6283–6304. https://doi.org/10.5194/acp-15-6283-2015
Miyazaki, Y., Sawano, M., & Kawamura, K. (2014). Low-molecular-weight hydroxyacids in
marine atmospheric aerosol: Evidence of a marine microbial origin. Biogeosciences, 11(16),
4407–4414. https://doi.org/10.5194/BG-11-4407-2014
Müller, J.-F., Stavrakou, T., Bauwens, M., Compernolle, S., & Peeters, J. (2018). Chemistry
and deposition in the Model of Atmospheric composition at Global and Regional scales using
Inversion Techniques for Trace gas Emissions (MAGRITTE v1.0). Part B. Dry deposition.
https://doi.org/10.5194/gmd-2018-317
Nah, T., Guo, H., Sullivan, A. P., Chen, Y., Tanner, D. J., Nenes, A., Russell, A., Lee Ng, N.,
Gregory Huey, L., & Weber, R. J. (2018). Characterization of aerosol composition, aerosol
acidity, and organic acid partitioning at an agriculturally intensive rural southeastern US site.
Atmospheric Chemistry and Physics, 18(15), 11471–11491. https://doi.org/10.5194/ACP-18-
639    11471-2018

Neeb, P., Sauer, F., Horie, O., & Moortgat, G. K. (1997). Formation of hydroxymethyl
hydroperoxide and formic acid in alkene ozonolysis in the presence of water vapour.
Atmospheric Environment, 31(10), 1417–1423. https://doi.org/10.1016/S1352-
643    2310(96)00322-6

Novakov, T., & Penner, J. E. (1993). Large contribution of organic aerosols to cloud-
condensation-nuclei concentrations. Nature 1993 365:6449, 365(6449), 823–826.
https://doi.org/10.1038/365823a0
Pan, X., Underwood, J. S., Xing, J.-H., Mang, S. A., & Nizkorodov, S. A. (2009).
Photodegradation of secondary organic aerosol generated from limonene oxidation by ozone
studied with chemical ionization mass spectrometry. Atmos. Chem. Phys, 9, 3851–3865.
https://doi.org/10.5194/acp-9-3851-2009
Paulot, F., Crounse, J. D., Kjaergaard, H. G., Kroll, J. H., Seinfeld, J. H., & Wennberg, P. O.
(2009). Isoprene photooxidation: New insights into the production of acids and organic nitrates.
Atmospheric Chemistry and Physics, 9(4), 1479–1501. https://doi.org/10.5194/acp-9-1479-
654    2009

Paulot, F., Wunch, D., Crounse, J. D., Toon, G. C., Millet, D. B., Decarlo, P. F., Vigouroux, C.,
Deutscher, N. M., Abad, G. G., Notholt, J., Warneke, T., Hannigan, J. W., Warneke, C., de Gouw,
J. A., Dunlea, E. J., de Mazière, M., Griffith, D. W. T., Bernath, P., Jimenez, J. L., & Wennberg,



P. O. (2011). Importance of secondary sources in the atmospheric budgets of formic and acetic
acids. Atmospheric Chemistry and Physics, 11(5), 1989–2013. https://doi.org/10.5194/acp-11-
660 1989-2011

Peng, X., Wang, T., Wang, W., Ravishankara, A. R., George, C., Xia, M., Cai, M., Li, Q.,
Salvador, C. M., Lau, C., Lyu, X., Poon, C. N., Mellouki, A., Mu, Y., Hallquist, M., Saiz-Lopez,
A., Guo, H., Herrmann, H., Yu, C., … Chen, J. (2022). Photodissociation of particulate nitrate
as a source of daytime tropospheric Cl2. Nature Communications, 13(1).
https://doi.org/10.1038/s41467-022-28383-9
Sanhueza, E., & Andreae, M. O. (1991). Emission of formic and acetic acids from tropical
Savanna soils. Geophysical Research Letters, 18(9), 1707–1710.
https://doi.org/10.1029/91GL01565
Saunders, S. M., Jenkin, M. E., Derwent, R. G., & Pilling, M. J. (2003). Protocol for the
development of the Master Chemical Mechanism, MCM v3 (Part A): Tropospheric degradation
of non-aromatic volatile organic compounds. Atmospheric Chemistry and Physics, 3(1), 161–
180. https://doi.org/10.5194/ACP-3-161-2003
Shaw, M. F., Sztáray, B., Whalley, L. K., Heard, D. E., Millet, D. B., Jordan, M. J. T., Osborn,
D. L., & Kable, S. H. (2018). Photo-tautomerization of acetaldehyde as a photochemical source
of formic acid in the troposphere. Nature Communications, 9(1), 1–7.
https://doi.org/10.1038/s41467-018-04824-2
Franco, B., Blumenstock, T., Cho, C. et al. Ubiquitous atmospheric production of organic acids
mediated by cloud droplets. Nature 593, 233–237 (2021). https://doi.org/10.1038/s41586-021-
03462-x
Stavrakou, T., Müller, J. F., Peeters, J., Razavi, A., Clarisse, L., Clerbaux, C., Coheur, P. F.,
Hurtmans, D., de Mazière, M., Vigouroux, C., Deutscher, N. M., Griffith, D. W. T., Jones, N.,
& Paton-Walsh, C. (2012). Satellite evidence for a large source of formic acid from boreal and
tropical forests. Nature Geoscience, 5(1), 26–30. https://doi.org/10.1038/ngeo1354
Su, T., Li, J., Li, C., Xiang, P., Lau, A. K. H., Guo, J., Yang, D., & Miao, Y. (2017). An
intercomparison of long-term planetary boundary layer heights retrieved from CALIPSO,
ground-based lidar, and radiosonde measurements over Hong Kong. Journal of Geophysical
Research: Atmospheres, 122(7), 3929–3943. https://doi.org/10.1002/2016JD025937
Toda, K., Yunoki, S., Yanaga, A., Takeuchi, M., Ohira, S.-I., & Dasgupta, P. K. (2014).
Formaldehyde Content of Atmospheric Aerosol. https://doi.org/10.1021/es500590e
Vet, R., Artz, R. S., Carou, S., Shaw, M., Ro, C. U., Aas, W., Baker, A., Bowersox, V. C.,
Dentener, F., Galy-Lacaux, C., Hou, A., Pienaar, J. J., Gillett, R., Forti, M. C., Gromov, S.,
Hara, H., Khodzher, T., Mahowald, N. M., Nickovic, S., … Reid, N. W. (2014). A global
assessment of precipitation chemistry and deposition of sulfur, nitrogen, sea salt, base cations,
organic acids, acidity and pH, and phosphorus. Atmospheric Environment, 93, 3–100.
https://doi.org/10.1016/J.ATMOSENV.2013.10.060





Walser, M. L., Park, J., Gomez, A. L., Russell, A. R., & Nizkorodov, S. A. (2007).
Photochemical Aging of Secondary Organic Aerosol Particles Generated from the Oxidation
of d-Limonene. https://doi.org/10.1021/jp066293l
Wolfe, G. M., Marvin, M. R., Roberts, S. J., Travis, K. R., & Liao, J. (2016). The framework
for 0-D atmospheric modeling (F0AM) v3.1. Geoscientific Model Development, 9(9), 3309–
3319. https://doi.org/10.5194/GMD-9-3309-2016
Xia, K., Tong, S., Zhang, Y., Tan, F., Chen, Y., Zhang, W., Guo, Y., Jing, B., Ge, M., Zhao, Y.,
Alamry, K. A., Marwani, H. M., & Wang, S. (2018). Heterogeneous Reaction of HCOOH on
NaCl Particles at Different Relative Humidities. Journal of Physical Chemistry A, 122(36),
7218–7226. https://doi.org/10.1021/ACS.JPCA.8B02790
Xia, M., Wang, T., Wang, Z., Chen, Y., Peng, X., Huo, Y., Wang, W., Yuan, Q., Jiang, Y., Guo,
H., Lau, C., Leung, K., Yu, A., & Lee, S. (2022). Pollution-Derived Br 2 Boosts Oxidation
Power of the Coastal Atmosphere. https://doi.org/10.1021/acs.est.2c02434
Xu, J., Chen, J., Shi, Y., Zhao, N., Qin, X., Yu, G., Liu, J., Lin, Y., Fu, Q., Weber, R. J., Lee, S.
H., Deng, C., & Huang, K. (2020). First Continuous Measurement of Gaseous and Particulate
Formic Acid in a Suburban Area of East China: Seasonality and Gas-Particle Partitioning. ACS
Earth and Space Chemistry, 4(2), 157–167.
https://doi.org/10.1021/acsearthspacechem.9b00210
Ye, C., Zhang, N., Gao, H., & Zhou, X. (2017). Photolysis of Particulate Nitrate as a Source of
HONO and NOx. Environmental Science and Technology, 51(12), 6849–6856.
https://doi.org/10.1021/acs.est.7b00387
Yokelson, R. J., Crounse, J. D., DeCarlo, P. F., Karl, T., Urbanski, S., Atlas, E., Campos, T.,
Shinozuka, Y., Kapustin, V., Clarke, A. D., Weinheimer, A., Knapp, D. J., Montzka, D. D.,
Holloway, J., Weibring, P., Flocke, F., Zheng, W., Toohey, D., Wennberg, P. O., … Shetter, R.
(2009). Emissions from biomass burning in the Yucatan. Atmospheric Chemistry and Physics,
9(15), 5785–5812. https://doi.org/10.5194/ACP-9-5785-2009
Yu, S. (2000). Role of organic acids (formic, acetic, pyruvic and oxalic) in the formation of
cloud condensation nuclei (CCN): a review. Atmospheric Research, 53(4), 185–217.
https://doi.org/10.1016/S0169-8095(00)00037-5
Yuan, B., Veres, P. R., Warneke, C., Roberts, J. M., Gilman, J. B., Koss, A., Edwards, P. M.,
Graus, M., Kuster, W. C., Li, S. M., Wild, R. J., Brown, S. S., Dubé, W. P., Lerner, B. M.,
Williams, E. J., Johnson, J. E., Quinn, P. K., Bates, T. S., Lefer, B., … de Gouw, J. A. (2015).
Investigation of secondary formation of formic acid: Urban environment vs. oil and gas
producing region. Atmospheric Chemistry and Physics, 15(4), 1975–1993.
https://doi.org/10.5194/acp-15-1975-2015
Zellner, R., Exner, M., & Herrmann, H. (1990). Absolute OH quantum yields in the laser
photolysis of nitrate, nitrite and dissolved H2O2 at 308 and 351 nm in the temperature range
278-353 K. Journal of Atmospheric Chemistry, 10(4), 411–425.



https://doi.org/10.1007/BF00115783
Zervas, E., Montagne, X., & Lahaye, J. (2001a). C1−C5 Organic Acid Emissions from an SI
Engine: Influence of Fuel and Air/Fuel Equivalence Ratio. Environmental Science and
Technology, 35(13), 2746–2751. https://doi.org/10.1021/ES000237V
Zervas, E., Montagne, X., & Lahaye, J. (2001b). Emission of specific pollutants from a
compression ignition engine. Influence of fuel hydrotreatment and fuel/air equivalence ratio.
Atmospheric Environment, 35(7), 1301–1306. https://doi.org/10.1016/S1352-2310(00)00390-

741 3

Zhang, R., Gen, M., Fu, T. M., & Chan, C. K. (2021). Production of Formate via Oxidation of
Glyoxal Promoted by Particulate Nitrate Photolysis. Environmental Science and Technology,
55(9), 5711–5720. https://doi.org/10.1021/acs.est.0c08199
Zhang, R., Suh, I., Zhao, J., Zhang, D., Fortner, E. C., Tie, X., Molina, L. T., & Molina, M. J.
(2004). Atmospheric New Particle Formation Enhanced by Organic Acids. New Series,
304(5676), 1487–1490. https://doi.org/10.1126/science.1095139
Zhu, C., Xiang, B., Zhu, L., & Cole, R. (2008). Determination of absorption cross sections of
surface-adsorbed HNO3 in the 290-330 nm region by Brewster angle cavity ring-down
spectroscopy. Chemical Physics Letters, 458(4–6), 373–377.
https://doi.org/10.1016/j.cplett.2008.04.125



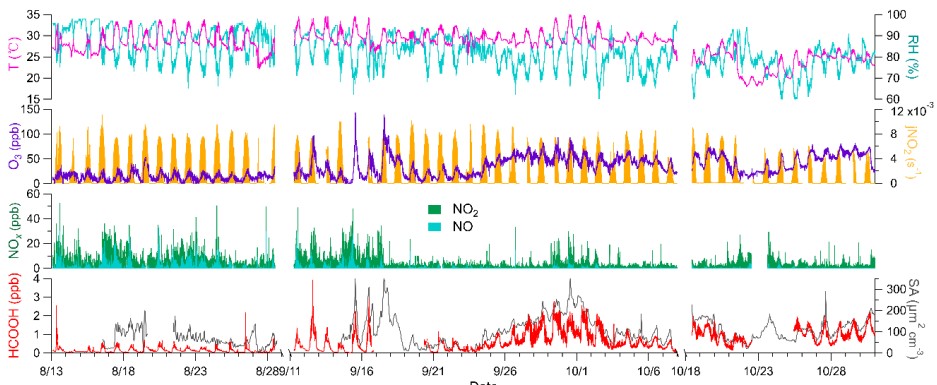

**Fig. 1** Times series of temperature (T); relative humidity (RH); nitrogen dioxide photolysis frequency (jNO₂); concentrations of ozone (O₃), nitrogen oxides (NOₓ), and formic acid (HCOOH); and surface area density (SA) during the field campaign. Data were not collected from August 29 to September 10 due to the persistently clean weather conditions, similar to those during the initial stage of the campaign. The gap in data collection from October 8 to October 17 was due to a rainstorm. Any other short gaps were caused by instrument maintenance.

**Table 1.** Summary of worldwide field-observed formic acid (HCOOH) concentrations

| Location | Type | Time | HCOOH (ppb) | Reference |
|---|---|---|---|---|
| Pasadena, USA | urban | 2010.06-07 | 2.0 ± 1.0 | (Yuan et al., 2015) |
| Kensington, London | urban | 2012.01-02 | 0.63 (winter) | (Bannan et al., 2017) |
| | background | 2012.07-08 | 1.33 (summer) | |
| Shanghai, China | suburban | 2017.06.18-12.23 | 2.08 ± 1.89 | (Xu et al., 2020) |
| Yorkville, USA | rural | 2016.08.15-10.13 | 1.17 ± 0.85 | (Nah et al., 2018) |
| North Pacific | marine | 2008.07.29-08.19 | 30 ± 39.8 ppt | (Miyazaki et al., 2014) |
| Pacific and Atlantic | marine | 2017.09-10 2018.04-05 | < 0.1 | (Chen et al., 2021) |
| Colorado, USA | forest | 2016.02.01-03.01 | 55 ± 57 ppt (winter) | (Fulgham et al., 2019) |
| | | 2016.04.15-05.15 | 30 ± 24 ppt (spring) | |
| | | 2016.07.15-08.15 | 1.2 ± 0.91 (summer) | |
| | | 2016.10.01-11.01 | 0.81 ± 0.48 (autumn) | |
| Alabama, USA | deciduous forest | 2013.06-07 | 2.5 (peak average daytime) | (Millet et al., 2015) |
| Hong Kong, China | coastal | 2021.08.13-10.31 | 0.58 ± 0.53 | This study |





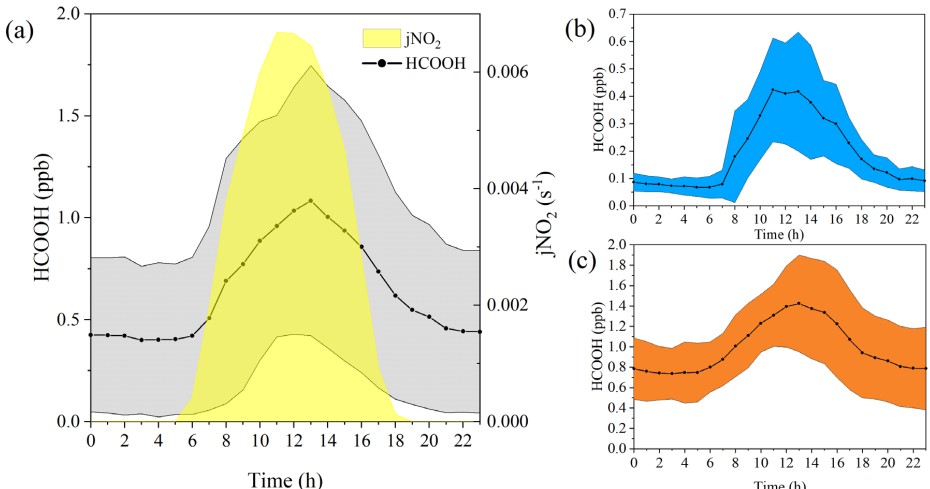

**Fig. 2** Diurnal variation in the concentrations of formic acid (HCOOH) in different periods. (a) Diurnal cycle of HCOOH concentrations and frequency of nitrogen dioxide photolysis ($jNO_2$) for the whole campaign; (b) and (c) diurnal cycle of HCOOH concentrations in marine and coastal air masses, respectively. The shading represents the standard deviations of the measurements.

**Table 2.** Pearson correlation coefficient ® matrix between the concentration of formic acid and other air pollutants, and related meteorological parameters, during three distinct periods.

| Parameter | Coastal | Haze | Marine | Parameter | Coastal | Haze | Marine |
|---|---|---|---|---|---|---|---|
| $jNO_2$ | 0.41 | 0.58 | 0.65 | Sa | 0.73 | 0.68 | -0.03 |
| T | -0.27 | 0.70 | 0.72 | $Sa \times NO_3^-$ | 0.85 | 0.56 | 0.15 |
| RH | -0.56 | -0.51 | -0.65 | $Sa \times O_3$ | 0.83 | 0.74 | 0.31 |
| $PM_1$ | 0.79 | 0.66 | 0.05 | $HNO_3$ | 0.75 | 0.59 | 0.69 |
| $PM_{2.5}$ | 0.69 | 0.63 | 0.19 | $Cl^-$ | -0.41 | -0.44 | 0.09 |
| $PM_{10}$ | 0.68 | 0.55 | 0.26 | $NO_3^-$ | 0.67 | -0.10 | 0.57 |
| HONO | -0.03 | 0.26 | -0.34 | $SO_4^{2-}$ | 0.66 | 0.65 | 0.10 |
| $CH_3COOH$ | 0.89 | 0.88 | -0.27 | $Na^+$ | -0.28 | -0.50 | 0.37 |
| NO | -0.12 | 0.44 | 0.13 | $NH_4^+$ | 0.72 | 0.64 | 0.24 |
| $NO_2$ | -0.24 | 0.36 | -0.39 | $K^+$ | 0.53 | 0.32 | 0.15 |
| $NO_x$ | -0.22 | 0.40 | -0.27 | $Mg^{2+}$ | -0.30 | -0.38 | 0.47 |
| $O_3$ | 0.69 | 0.65 | 0.68 | $Ca^{2+}$ | -0.11 | 0.09 | 0.04 |
| $SO_2$ | 0.64 | 0.66 | 0.41 | HCl | 0.18 | 0.51 | 0.55 |
| CO | 0.63 | 0.51 | 0.13 | isoprene | 0.03 | 0.61 | 0.63 |
| $NH_3$ | 0.37 | 0.46 | 0.16 | benzene | 0.63 | 0.55 | 0.05 |


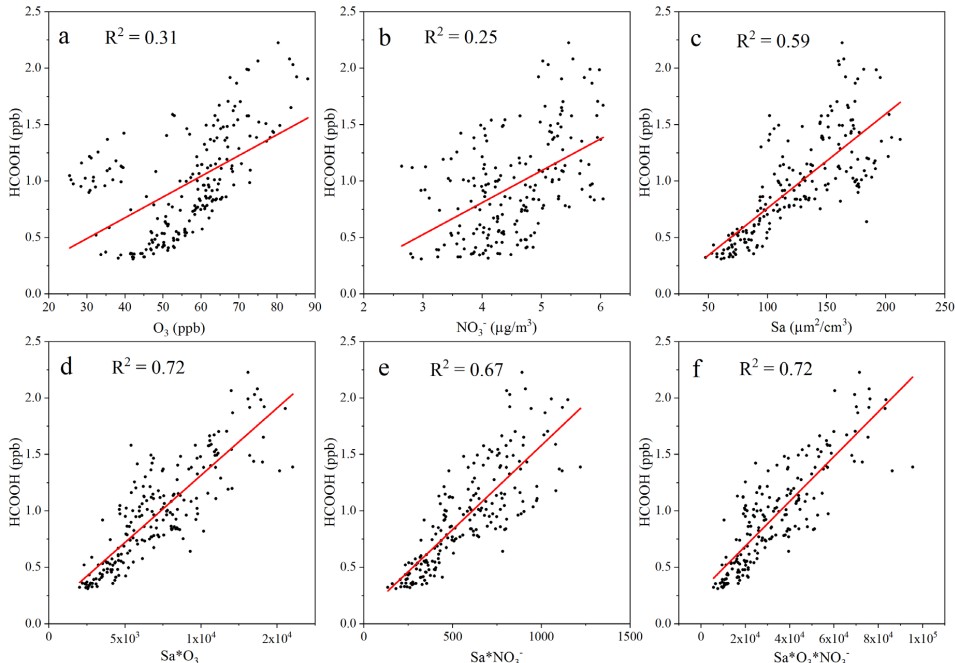

**Fig. 3** Scatter plot of the concentration of formic acid (HCOOH) and (a) the concentration of ozone ($O_3$); (b) the mass concentration of nitrate ions ($NO_3^-$) in $PM_{2.5}$; (c) the surface area density (Sa) of $PM_1$ ($\mu m^2$ $cm^{-3}$); (d) the product of Sa and the concentration of $O_3$; (e) the product of Sa and the concentration of $NO_3^-$; and (f) the product of Sa, the concentration of $O_3$, and the concentration of $NO_3^-$ in coastal air masses.





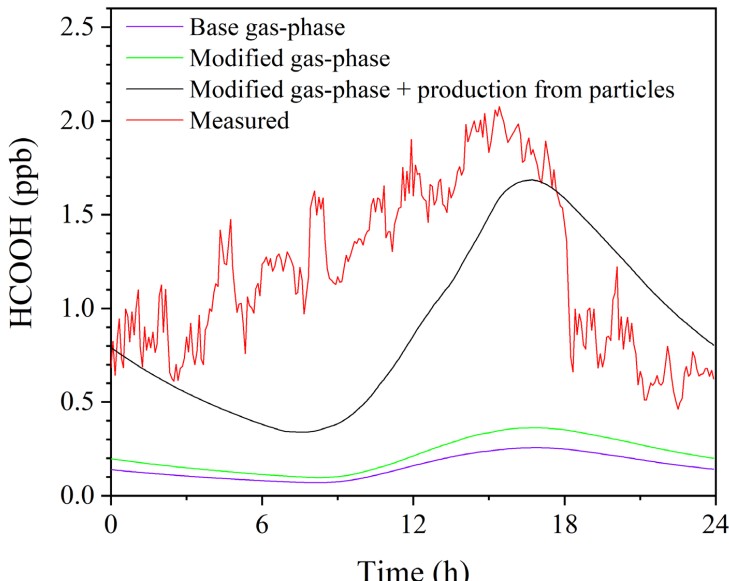

**Fig. 4** Variations in the concentrations of formic acid (HCOOH) on 28 September 2021.

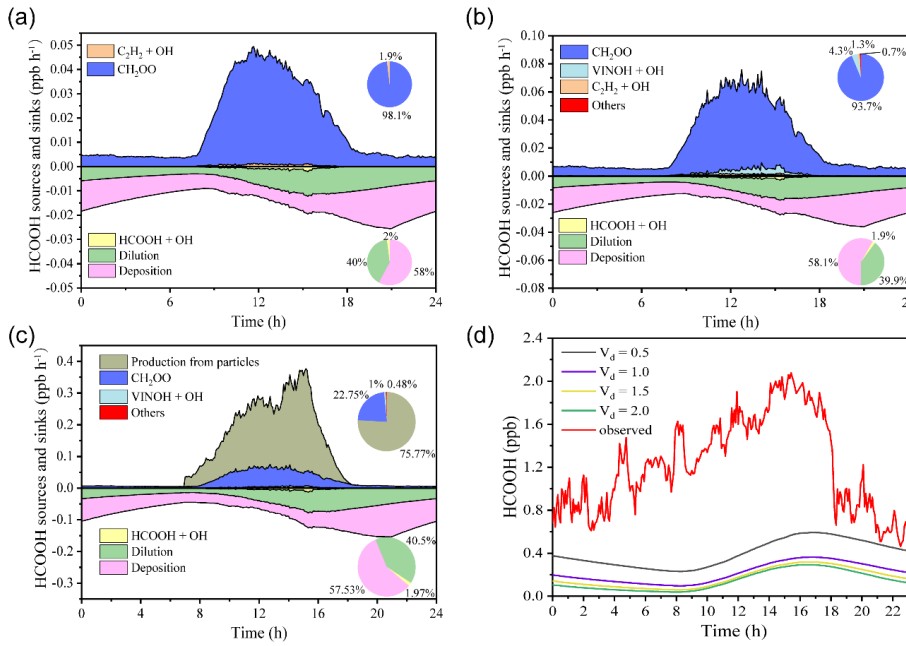

**Fig. 5** Model-calculated profiles of sources and sinks of formic acid (HCOOH) on 28 September 2021 for (a) the base case; (b) the modified case; and (c) the case including the production from particles. Upper right inset: the contribution from various sources to HCOOH concentrations. Bottom right inset: the contribution from different sinks to HCOOH



concentrations. $CH_2OO$ = formaldehyde oxide, a Criegee intermediate (biradical); VINOH =
vinyl alcohol. d) Model-predicted concentrations of HCOOH based on various deposition
velocities ($V_d$).

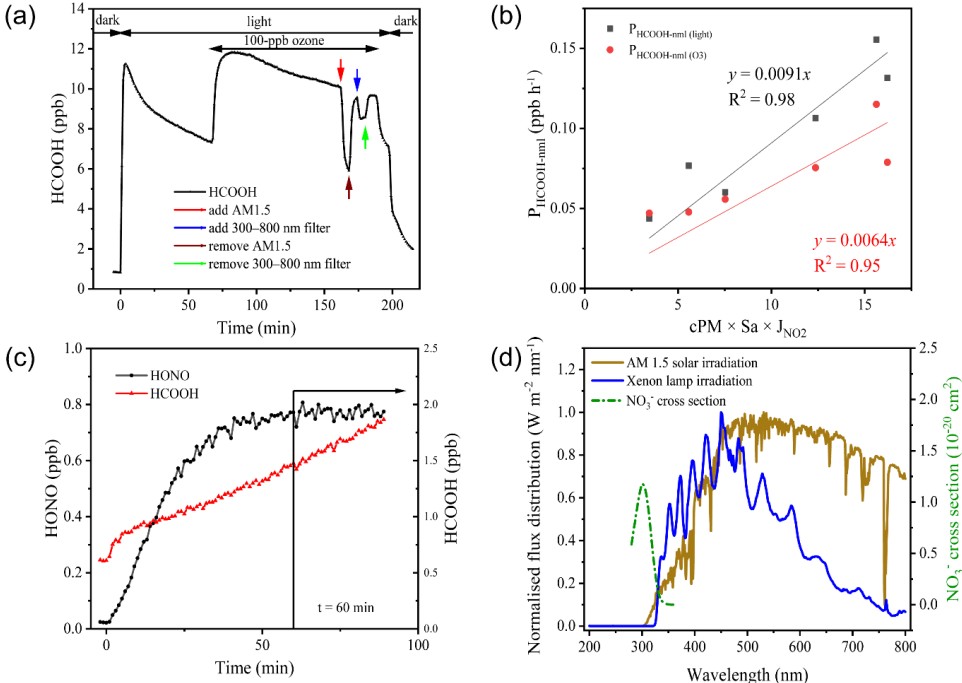

**Fig. 6** Results of the irradiation experiments. (a) Typical variation in formic acid (HCOOH)
concentrations during irradiation (in aerosols collected on 2 November 2020). AM 1.5 and
300–800 nm filters were added after the addition of 100-ppb ozone. (b) The correlation between
$P_{HCOOH-nml}$ and 2.5-µm particulate matter ($PM_{2.5}$) concentration (cPM) × surface area (Sa) ×
nitrogen dioxide photolytic frequency ($jNO_2$). (c) Variations in the concentrations of HCOOH
and nitrous acid (HONO) as a function of time after illumination of an aqueous solution of
formaldehyde and sodium nitrite at pH = 2.7. The vertical black line indicates the time at which
HONO stabilised. (d) Comparison of the irradiation spectrum of the xenon lamp used in this
study and standard air mass 1.5 solar irradiation (AM 1.5).



**Table 3** Summary of formic acid (HCOOH) concentrations and production rates observed in
chamber experiments and normalised HCOOH production rates in ambient air under light and
light + ozone ($O_3$) conditions, respectively. $HCOOH_{(O3)}$ denotes the increased concentration of
HCOOH after the addition of 100 ppb $O_3$.

| Date | $HCOOH_{(light)}$ (ppt) | $HCOOH_{(O3)}$ (ppt) | $P_{HCOOH(light)}$ (ppt s$^{-1}$) | $P_{HCOOH(O3)}$ (ppt s$^{-1}$) | $P_{HCOOH-nml\ (light)}$ (ppb h$^{-1}$) | $P_{HCOOH-nml\ (O3)}$ (ppb h$^{-1}$) |
|---|---|---|---|---|---|---|
| 2020.10.07 | 8420.2 | 4670.0 | 299.4 | 166.0 | 1.70E-01 | 1.33E-01 |
| 2020.10.08 | 6787.7 | 2899.0 | 241.3 | 103.1 | 1.31E-01 | 7.89E-02 |
| 2020.10.26 | 4660.9 | 3077.5 | 165.7 | 109.4 | 6.01E-02 | 5.57E-02 |
| 2020.11.02 | 6656.3 | 3507.6 | 236.7 | 124.7 | 1.55E-01 | 1.15E-01 |
| 2020.11.03 | 4490.8 | 2266.7 | 159.7 | 80.6 | 1.06E-01 | 7.55E-02 |
| 2020.11.04 | 4943.1 | 2191.5 | 175.8 | 77.9 | 7.67E-02 | 4.77E-02 |
| 2020.11.05 | 3088.0 | 2368.6 | 109.8 | 84.2 | 4.37E-02 | 4.71E-02 |
| Average | 5578.1 | 2997.3 | 198.3 | 106.6 | 1.06E-01 | 7.89E-02 |
