# Peer review of "Photochemical aging of aerosols contributes significantly to the"

_EGUsphere, 2023_

## Author Comment (AC1)

Response to reviewer comments on "Photochemical aging of aerosols contributes significantly to the production of atmospheric formic acid" by Yifan Jiang et al.

Comments from the reviewers are shown in black *Italic* font. Response from the authors is shown in black regular font. Revisions are shown in blue regular font. The line numbers provided here refer to the ones in the revised manuscript.

*Reviewer 1 Comments:*
*This paper examines the sources of formic acid (HCOOH) in the somewhat polluted marine boundary layer. The authors have an excellent site on the coast near Hong Kong that samples a range of polluted conditions from coastal to marine. The mix of air masses was a bit unusual as the marine was highly polluted in NOx with low O3, and the land(coastal) air had high O3 and biogenics. They identify aerosols as a key co-existing species that is related to high formic acid levels. They track the observed air parcels with HYSPLIT back trajectories. They then pursue laboratory-chamber studies to quantify the net production of HCOOH from different aerosols and follow up with a box model study of the kinetics involved. Their conclusions that a major source of HCOOH comes from photochemical aging of organic compounds in aerosols (particularly nitrate-containing aerosols) is indisputable. They note that inclusion of this additional source would reduce some (all) of the model-measurement discrepancy in global models. The paper is very clearly written; and from my fast read-through, I did not find any typos. Altogether, impressive.*

Response: Thank you for your encouraging comments. Below please find our point-to-point response and revisions of the manuscript.

*1. "191.1 ± 167.2 ppt in marine air masses" – the 4th decimal place is unnecessary and only clutters up the major numbers: 191 ±167.*

Response: We agree that the fourth decimal place is unnecessary. We have revised as the referee suggested.

Revision in the main text:
Abstract:
Line 17-18: The average concentrations of HCOOH were $191 \pm 167$ ppt in marine air masses and $996 \pm 433$ ppt in coastal air masses.
Results Section 3.1:
Line 289-296: The average HCOOH concentration in marine air masses was $191 \pm 167$ ppt; this was higher than those over the remote ocean, due to local emission sources, but significantly lower than those in urban environments (Table 1). In contrast, the ambient HCOOH concentrations in coastal air masses were substantially higher, averaging $996 \pm 433$ ppt, comparable with other measurements at rural or urban background sites. During the haze period, the concentrations of HCOOH displayed a pattern similar to the concentrations of $O_3$, with the daytime peak concentration

increasing from 674 to 2790 ppt.

*2. The lack of atmospheric HCOOH sources in models is duly noted, but do the aerosols in the dominant remote marine atmosphere have the organics and nitrates to generate the missing source? Can the authors assess/speculate on this based on published data, e.g., from ATom. I do not expect them to analyze other observations, but they can comment on whether the aerosols observed over the Pacific would likely produce ~0.1 ppb/hr as at their site.*

Response:
Following the referee's suggestion, we estimated the production rates of HCOOH from aerosols using the Atmospheric Tomography Mission (ATom) data from April to May 2018 (Wofsy et al., 2021). The selected observation data were obtained within the Pacific Ocean with sampling heights below 5 km. The mean values of mass concentration of $PM_1$, the surface area density of $PM_1$, the photolysis frequency of $NO_2$ and the mixing ratio of $O_3$ were 1 µg $cm^{-3}$, 26 $µm^2$ $cm^{-3}$, 0.01 $s^{-1}$ and 24.2 ppb, respectively. The calculated mean production rate of HCOOH from aerosols ($P_{HCOOH-a}$) was 6.6 ppt $h^{-1}$. Considering that the proportion of nitrate in $PM_1$ was only 3%, which is significantly lower than that at our site (24.3%), the actual $P_{HCOOH-a}$ is expected to be even lower. The small $P_{HCOOH-a}$ in the remote marine atmosphere is reasonable given the low concentration of HCOOH observed over the Pacific (mean: 10.5 ppt; maximum: 85.6 ppt). In addition, a previous study showed small discrepancy between observed and modelled results (without considering aerosol aging) in remote clean air masses (Chen et al., 2021).

We also assessed the HCOOH production when the remote marine boundary layer is affected by fire plumes by selecting data obtained at heights below 5 km and HCOOH concentrations higher than 1 ppb. The mean values of the mass concentration of $PM_1$, the surface area density of $PM_1$, the photolysis frequency of $NO_2$ and the mixing ratio of $O_3$ were 4.9 µg $cm^{-3}$, 92.6 $µm^2$ $cm^{-3}$, 0.011 $s^{-1}$ and 45.5 ppb, respectively. The resulting $P_{HCOOH-a}$ was 84.5 ppt $h^{-1}$, significantly higher than that observed in remote marine atmosphere. After consideration of the low proportion of nitrate in $PM_1$ (5.1%), $P_{HCOOH-a}$ was 17.7 ppt $h^{-1}$ assuming a positive linear correlation between $P_{HCOOH-a}$ and nitrate concentration. This corresponds to a rate of 2390%/year that organic aerosol (OA) mass (3 µg $cm^{-3}$) is photochemically converted to HCOOH, which is equivalent to a carbon-based HCOOH yield of ~3.8−38% over 1−10 days of aging, close to that required to account for the ATom observations (16−37%) in aged fire air masses (Chen et al., 2021). Therefore, the photochemical aging of aerosols potentially explains a substantial portion of missing HCOOH sources in aged fire plumes. We added discussions above in the main text and the calculation details are shown in SI.

Revision in the main text:
Results Section 3.3:
Line 456-464: To evaluate the role of aerosol photochemical aging on HCOOH production in a broader context, we also examined the HCOOH data over the remote

marine boundary layer obtained from the Atmospheric Tomography Mission (ATom) aircraft campaign which was conducted around the globe during April-May 2018 (Wofsy et al., 2021) (Text S5). Our results show that the photochemical aging of aerosols was insignificant in remote ocean areas due to the low PM and nitrate concentrations found there. However, when these regions are affected by aged fire plumes containing higher levels of organics and nitrate, photochemical aging of aerosols accounts for the substantial sources of HCOOH. These results suggest the photochemical aging appears to be important in relatively polluted atmospheres.

Revision in SI:

Line 100-128:

**Text S5.** Evaluation methods of HCOOH production from photochemical aging of aerosols using the Atmospheric Tomography Mission (ATom) data.

We first assessed the production of HCOOH from aerosols over the Pacific as an illustrative case of the remote marine boundary layer. The selected observation data were obtained within the Pacific Ocean with sampling heights below 5 km. The mean values of mass concentration of $PM_1$, the surface area density of $PM_1$, the photolysis frequency of $NO_2$ and the mixing ratio of $O_3$ were 1 $\mu g\ cm^{-3}$, 26 $\mu m^2\ cm^{-3}$, 0.01 $s^{-1}$ and 24.2 ppb, respectively. The calculated mean production rate of HCOOH from aerosols ($P_{HCOOH-a}$) was 6.6 ppt $h^{-1}$. Considering that the proportion of nitrate in $PM_1$ was only 3%, which is significantly lower than that at our site (24.3%), the actual $P_{HCOOH-a}$ is expected to be even lower. The small $P_{HCOOH-a}$ in the remote marine atmosphere is reasonable given the low concentration of HCOOH observed over the Pacific (mean: 10.5 ppt; maximum: 85.6 ppt). In addition, a previous study showed small discrepancy between observed and modeled results in remote clean air masses in remote clean air masses (Chen et al., 2021).

We also assessed the HCOOH production when the remote marine boundary layer is affected by fire plumes by selecting data obtained at heights below 5 km and HCOOH concentrations higher than 1 ppb. The mean values of the mass concentration of $PM_1$, the surface area density of $PM_1$, the photolysis frequency of $NO_2$ and the mixing ratio of $O_3$ were 4.9 $\mu g\ cm^{-3}$, 92.6 $\mu m^2\ cm^{-3}$, 0.011 $s^{-1}$ and 45.5 ppb, respectively. The resulting $P_{HCOOH-a}$ was 84.5 ppt $h^{-1}$, significantly higher than that observed in remote marine atmosphere. After consideration of the low proportion of nitrate in $PM_1$ (5.1%), $P_{HCOOH-a}$ was 17.7 ppt $h^{-1}$ assuming a positive linear correlation between $P_{HCOOH-a}$ and nitrate concentration. This corresponds to a rate of 2390%/year that organic aerosol (OA) mass (3 $\mu g\ cm^{-3}$) is photochemically converted to HCOOH, which is equivalent to a carbon-based HCOOH yield of ~3.8−38% over 1−10 days of aging, close to that required to account for the ATom observations (16−37%) in aged fire air masses (Chen et al., 2021).

**References mentioned in author's response**:

Chen, X., Millet, D. B., Neuman, J. A., Veres, P. R., Ray, E. A., Commane, R., Daube, B. C., McKain, K., Schwarz, J. P., Katich, J. M., Froyd, K. D., Schill, G. P., Kim, M. J., Crounse, J. D., Allen, H.

M., Apel, E. C., Hornbrook, R. S., Blake, D. R., Nault, B. A., Campuzano-Jost, P., Jimenez, J. L., and Dibb, J. E.: HCOOH in the Remote Atmosphere: Constraints from Atmospheric Tomography (ATom) Airborne Observations, ACS Earth Space Chem, 5, 1436–1454, https://doi.org/10.1021/acsearthspacechem.1c00049, 2021.

Wofsy, S.C., S. Afshar, H.M. Allen, E.C. Apel, E.C. Asher, B. Barletta, J. Bent, H. Bian, B.C. Biggs, D.R. Blake, N. Blake, I. Bourgeois, C.A. Brock, W.H. Brune, J.W. Budney, T.P. Bui, A. Butler, P. Campuzano-Jost, C.S. Chang, M. Chin, R. Commane, G. Correa, J.D. Crounse, P. D. Cullis, B.C. Daube, D.A. Day, J.M. Dean-Day, J.E. Dibb, J.P. DiGangi, G.S. Diskin, M. Dollner, J.W. Elkins, F. Erdesz, A.M. Fiore, C.M. Flynn, K.D. Froyd, D.W. Gesler, S.R. Hall, T.F. Hanisco, R.A. Hannun, A.J. Hills, E.J. Hintsa, A. Hoffman, R.S. Hornbrook, L.G. Huey, S. Hughes, J.L. Jimenez, B.J. Johnson, J.M. Katich, R.F. Keeling, M.J. Kim, A. Kupc, L.R. Lait, K. McKain, R.J. Mclaughlin, S. Meinardi, D.O. Miller, S.A. Montzka, F.L. Moore, E.J. Morgan, D.M. Murphy, L.T. Murray, B.A. Nault, J.A. Neuman, P.A. Newman, J.M. Nicely, X. Pan, W. Paplawsky, J. Peischl, M.J. Prather, D.J. Price, E.A. Ray, J.M. Reeves, M. Richardson, A.W. Rollins, K.H. Rosenlof, T.B. Ryerson, E. Scheuer, G.P. Schill, J.C. Schroder, J.P. Schwarz, J.M. St.Clair, S.D. Steenrod, B.B. Stephens, S.A. Strode, C. Sweeney, D. Tanner, A.P. Teng, A.B. Thames, C.R. Thompson, K. Ullmann, P.R. Veres, N.L. Wagner, A. Watt, R. Weber, B.B. Weinzierl, P.O. Wennberg, C.J. Williamson, J.C. Wilson, G.M. Wolfe, C.T. Woods, L.H. Zeng, and N. Vieznor.: ATom: Merged Atmospheric Chemistry, Trace Gases, and Aerosols, Version 2 (Version 2.0), ORNL Distributed Active Archive Center, https://doi.org/10.3334/ornldaac/1925, 2021.

---

## Author Comment (AC2)

Response to reviewer comments on "Photochemical aging of aerosols contributes significantly to the production of atmospheric formic acid" by Yifan Jiang et al.

Comments from the reviewers are shown in black *Italic* font. Response from the authors is shown in black regular font. Revisions are shown in blue regular font. The line numbers provided here refer to the ones in the revised manuscript.

*Reviewer 2 Comments:*
*In their study, Jiang et al. estimate the production of formic acid from photochemical aging of ambient particles by combining observations, laboratory experiments, and HYSPLIT and box modelling. Their proposed HCOOH production pathway can explain some of the major underestimation in their performed box model studies. Overall, the manuscript is well written but further sensitivity studies are necessary before the manuscript can be published in ACP.*

Response: Thank you for the overall positive comment and valuable suggestions for improvement. We have conducted additional sensitivity studies to further enhance our research, as you suggested.

*Major comments:*

*My major concern is related to the box modelling performed. Overall, the box model setup used is in part too simplified and some assumptions by the authors influence the predicted HCOOH concentration. For all box model simulations performed, the model fails to properly predict the diurnal cycle of HCOOH. From midnight to about 2pm (Fig. 4), a steady increase in HCOOH is observed but the model predicts a decrease until sunrise. After sunset, a sharp decrease in HCOOH is observed, which the model fails to reproduce. The authors only discuss uncertainties in the deposition velocities, which do not resolve issues in reproducing the diurnal cycle. I suggest performing further modelling sensitivity simulations to challenge some of the assumptions made by the authors. These simulations should at least be concerned with:*

Response: We agree with the referee that the box model didn't reproduce the observed diurnal cycle of HCOOH, in part because it only considered local chemistry but ignored transport of HCOOH which has a moderate lifetime of 2-4 days. In addition, the HCOOH concentration is also affected by physical processes (e.g., dilution and deposition). Thus, we mainly focused on the enhancement effect of photochemical aging of aerosols on the production rate of HCOOH. For simulations on HCOOH concentrations, we also performed additional sensitivity tests. The modelling portion has gone through major revision. Below please find our point-by-point responses and revisions.

*1. The authors acknowledge that there are direct biogenic and anthropogenic emissions of HCOOH in the vicinity of the station and in line 280, they acknowledge a marine and anthropogenic influence. However, these influences are ignored in the box model. It should be checked if influx from other regions might explain some of the discrepancies in the early morning.*

Response: We agree with the reviewer to address the biogenic and anthropogenic emissions as sources of HCOOH. Our observation site is located in a vehicle-restricted area with no obvious anthropogenic emissions nearby, except for ships emissions in the open ocean, with the main shipping lanes located 8 kilometres to the south or southwest. The significance of ship emissions as a source of HCOOH during the modelling period should be minor for the following reasons: 1. the HCOOH concentrations were relatively low in marine air masses (with high $NO_x$) 2. the modelling period was dominated by continental outflows which are unfavourable for the transportation of HCOOH from ship emissions to our site.

We estimated the biogenic emissions using the algorithm of The Model of Emissions of Gases and Aerosols from Nature version 2.1 (MEGAN v2.1) (Guenther et al., 2012). The biogenic emission contributed about 34% to the HCOOH production. It is worth noting that this approach may introduce some uncertainties in estimating biogenic emission at a specific location. Details of the estimation of biogenic emissions are now shown in SI Text S3.

Revision in the main text:
Methods Section 2.1:
Line 109-113: During the study period, the air quality of this coastal site was initially predominantly influenced by marine air masses from the South China Sea and subsequently by the coastal air masses transporting regional anthropogenic pollution from East China. The site was also affected by biogenic emissions from local vegetation and ship emissions transported mainly from about 8 kilometres away.
Methods Section 2.4:
Line 257-265: The local sources of HCOOH at this site mainly consist of ship and biogenic emissions. The box model used in this study did not account for the contribution of ship emissions since the modelling period was dominated by continental outflows which is unfavourable for the transportation of HCOOH from ship emissions to our site as evidenced by the relatively low concentrations of $NO_x$ in the modelling period. To estimate the biogenic emissions, we used the algorithm of the Model of Emissions of Gases and Aerosols from Nature version 2.1 (MEGAN v2.1) (Guenther et al., 2012), assuming instantaneous dilution into the whole box. It is worth noting that this approach may introduce some uncertainties in estimating biogenic emission at a specific location. The specific parameters used can be found in Text S3.
Results Section 3.3:
Line 421-423: The biogenic emissions (S4) also played an important role, contributing to 34.4% of the total production (Fig. 5d).

Revision in SI:
Line 73-82: The biogenic emissions of HCOOH were calculated using the exponential temperature dependence algorithm of the Model of Emissions of Gases and Aerosols

from Nature version 2.1 (MEGAN v2.1) (Guenther et al., 2012), as shown in (Eq. (2)).

$$E = \varepsilon \text{LAI} \gamma_P \gamma_T \quad (2)$$

where E is the biogenic emission of HCOOH ($\mu g \, m^{-2} \, h^{-1}$); $\varepsilon$ is the emission factor under standard environmental conditions (30 $\mu g \, m^{-2} \, h^{-1}$, Paulot et al. 2011); LAI is the leaf area index (3.65 $m^2 \, m^{-2}$, Myneni et al., 2021); $\gamma_P$ and $\gamma_T$ are the emission activity factors accounting for variability in light and temperature. In particular, $\gamma_P$ was calculated using the PCEEA algorithm described by Guenther et al. (2006) and $\gamma_T$ was calculated following Paulot et al. (2011).

*2. I checked out METEOSAT images and there were clouds reported for that day. Even though there was no precipitation, by vertical mixing, the production pathway proposed by Franco et al. could still be important and its contribution should be tested in the box model.*

Response: We agree that there is potential in-cloud production of HCOOH followed by its transport to the surface. We acknowledge that our zero-dimensional box model is unable to assess the contribution of cloud-related chemistry to HCOOH observed at our site. In the revised version, we mainly discuss gas-phase and aerosol aging processes. We added the following text in the revised manuscript.

Revision in the main text:
Methods Section 2.4:
Line 270-273: However, the model used in this study was unable to account for the downward transport of HCOOH produced in clouds through a newly proposed multiphase pathway (Franco et al., 2021) due to its inability to assess the contribution of vertical mixing and aqueous phase chemistry.

*3. After sunset, there is a sharp decrease in the observed formic acid. I suspect that changes in the transport pattern might play a role, due to the station being so close to the ocean. I checked the raw data provided by the authors and noticed that there was a substantial change in wind direction and speed in the afternoon. I suggest performing a HYSPLIT analysis for this day and investigating if the change in transport pattern might explain the sharp decrease. In the model, this in-/outflow could be added as an additional production-/loss term. A similar approach could be used for point 1.*

Response: We performed a 72h HYSPLIT analysis for that day and found a sudden change from coastal air masses to marine air masses at 18:00, which caused the sharp decrease in HCOOH concentrations at that time. In the revised version, we changed analysis of diurnal profile on 28 September to campaign-averaged diurnal profile, following the referee's suggestion (see the comment below). The Sept case is no longer subject to detail budget analysis. Regarding the model's failure to predict the sharp decrease in concentration, we suspect that this can be attributed to the oversimplification of a constant physical loss rate in capturing the complex physical processes. To address this issue, we applied a bimodal physical loss rate to reflect variations in mixing rates with background air at different times of the day and found it

predicted HCOOH concentrations better than before. Please find details in the Revision in the main text.

Revision in the main text:
Results Section 3.3:
Line 443-447: Therefore, to account for the complex physical processes, we employed a bimodal physical loss rate due to vertical dilution that varied with time of day ($1/21,600$ $s^{-1}$ in daytime and a much smaller value of $1/518,400$ $s^{-1}$ at night), as suggested by Yuan et al. (2015). It is clear that the model performed better in predicting the diurnal pattern using the bimodal physical loss rate compared to a constant value (Fig. 6).

*The authors limit their modelling to two separate days but limit their main analysis to only one day. Why not performing longer simulations? From Figure 1, we clearly see a different behavior on the next day (29 September 2021). On this day, a sharp increase in formic acid is observed early in the morning. Is the production pathway proposed in this study able to reproduce this behavior? I strongly suggest performing long term simulations for which a high variability in formic acid is observed. From Fig. 1 and the raw data provided there should be sufficient data to perform this analysis from 9 September to 6 October.*

Response: Our previous analysis focused on September 28th due to the limited continuous OVOCs measurements on other days. Following the referee's suggestion, we have performed longer simulations, from 24 September to 7 October by using modelled OVOCs and adoption of bimodal physical loss rates. We excluded data from 9 to 23 September due to missing data of either surface area density or HCOOH concentrations. We also replaced the single day analysis with campaign-averaged diurnal profile which is more representative. We found that the modelled results using observed and modelled OVOCs values are almost the same. After applying a bimodal physical loss rate, the model predicts well in the continuous simulation of two weeks, except for two days with high PM concentrations but low nitrate proportion. Further details can be found in the revised main text and SI.

Substantial revisions have been made on the model related parts in the main text:
Methods Section 2.4:
Line 251-254: Wet deposition was not considered as there was no rainfall except on 3 October and 7 October. We simulated the averaged diurnal cycle for the whole campaign with field-observed relevant species constrained hourly in the model. Simulations were also performed daily for a 2-week period, from 24 September to 7 October. The details of input data are described in Text S3.
Results Section 3.3:
Line 403-411: We next use a box model (see Methods 2.4) to evaluate the production and loss of HCOOH in four scenarios (Table 4) which include the default MCM mechanism (S1), modified with gas-phase reactions following Yuan et al. (2015) (S2), further addition of the photochemical aging source (S3), and further adding of a biogenic source (S4). Figure 5 presents a comprehensive analysis of HCOOH budget

of campaign-averaged diurnal profile. In the base case (Default MCM), the highest net production rate of HCOOH ($P_{HCOOH-net}$) is 0.018 ppb h$^{-1}$, significantly lower than the observed average rate of increase in HCOOH concentrations of 0.095 ppb h$^{-1}$ from 6 am to 1 pm. Despite an increase of $P_{HCOOH-net}$ to 0.031 ppb h$^{-1}$ in modified case, the modelled value still lower than the observed rate of change.

Line 417-436:

Incorporating the photochemical production of HCOOH from particles into the F0AM model (S3) resulted in substantial improvements in predictions. The peak $P_{HCOOH-net}$ increased to 0.073 ppb h$^{-1}$, which is more than double that of the modified case. Among the secondary production mechanisms considered, the production of HCOOH from particles was found to be the largest source, accounting for 52.1% of the secondary production (Fig. 5c). The biogenic emissions (S4) also played an important role, contributing to 34.4% of the total production (Fig. 5d). After considering all sources, including primary emissions and secondary productions, the modelled $P_{HCOOH-net}$ (0.094 ppb h$^{-1}$) was much closer to the observed increase rate of HCOOH (0.095 ppb h$^{-1}$).

Figure 6 presents the observed and modelled concentrations of HCOOH using different mechanisms for the averaged diurnal profile throughout the whole campaign. It is evident that the predicted HCOOH concentration increased substantially after incorporating the productions from the photochemical aging of aerosols. However, the modelled HCOOH concentration is still lower than the observed value in Scenario 4, where all sources are included. The discrepancy may be explained by the inadequate treatment of physical processes in the box model, such as deposition, convection, and advection. The primary loss of HCOOH is via deposition, owing to its high solubility in water. To account for uncertainty in the deposition velocity ($V_d$) of HCOOH, we conducted a sensitivity test of HCOOH production to various $V_d$ values in Scenario 4 (Fig. S6). The results revealed that a smaller deposition velocity results in higher modelled HCOOH concentrations. The daytime peak concentration increased by about 20% when $V_d$ decreased from 1.00 to 0.50 cm s$^{-1}$.

Line 448-455:

After applying a bimodal physical loss rate, the model also predicts better in the continuous simulation of two weeks (Fig. S7), except for 30 September and 1 October. The observed HCOOH concentrations on these two days were significantly lower than the modelled values due to a lower nitrate proportion (13.5%) on 30 September compared to other days during the model period (22.2%). Therefore, the simplified parameterization using PM$_{2.5}$ may overestimate the production of HCOOH from photochemical aging of aerosols in areas with high PM concentrations but a low nitrate proportion. An improved parameterization using the concentrations of nitrate and organics should be developed in future studies.

Line 827-835:

[Figure]

**Fig. 5** Model-calculated profiles of sources and sinks of formic acid (HCOOH) on averaged diurnal profile during the whole campaign in four scenarios described in Table 4. Upper right inset: the contribution from various sources to HCOOH concentrations. Bottom right inset: the contribution from different sinks to HCOOH concentrations. $CH_2OO$ = formaldehyde oxide, a Criegee intermediate (biradical); VINOH = vinyl alcohol.

[Figure]

**Fig. 6** Comparison of measured and modelled diurnal profiles of HCOOH during the whole campaign.

Line 836-837:

**Table 4** The mechanisms included in different model scenarios.

| scenario | Default MCM | Modified MCM | Particle-phase pathway | Biogenic emissions |
|---|---|---|---|---|
| 1 | √ | | | |
| 2 | | √ | | |
| 3 | | √ | √ | |
| 4 | | √ | √ | √ |

Revision in SI:

Line 52-70:

**Text S3.** Additional setting information on the box model simulation.

The data of trace gases (including $O_3$, NO, $NO_2$, CO and $SO_2$) and data obtained by ToF-CIMS (including HONO and $N_2O_5$) were averaged to one-hour resolution. For missing values of VOCs species, linear interpolations were applied. The missing $J_{NO2}$ data was filled in using the data first calculated by the Tropospheric Ultraviolet and Visible (TUV) Radiation Model (https://www2.acom.ucar.edu/modeling/tropospheric-ultraviolet-and-visible-tuv-radiation-model) and then scaled using measured $J_{NO2}$. Since the OVOCs data was not available during most of the modelling period, we run the model using measured VOCs concentration for three replicates to stabilise the

intermediate species it generated and assumed the output concentrations of OVOCs were equal to the ambient values. We compared the simulated HCOOH concentration on 28 September where the ambient measured OVOCs data was available using modelled and measured values (Fig. S8). The uncertainty caused by using modelled OVOCs values is negligible. The methane concentration was assumed to remain constant (2000 ppb; Peng et al., 2022), due to a lack of measurement data. The variable "ModelOptions.EndPointsOnly" was set to "1" because we only want the last point of each step. The "ModelOptions.LinkSteps" were set to "1" so that non-constrained species are carried over between steps. The variable "ModelOptions.IntTime" was set to "3600" meaning that the integration time for each step was 3600s.

Line 145-156:

[Figure]

**Figure S5.** The sources of $CH_2OO$ in the modified case on averaged diurnal profile during the whole campaign.

[Figure]

**Figure S6.** The sensitivity test of deposition velocity on averaged diurnal profile during the whole campaign in Scenario 4.

[Figure]

**Figure S7.** The model results of the simulation of two weeks using bimodal physical loss rates.

[Figure]

**Figure S8.** Comparison of simulated HCOOH concentration on 28 September using measured and simulated OVOCs values.

*Detailed comments:*

*Line 54-56: The recently updated formic acid budget presented in Franco et al. 2021 needs to be discussed in this context. Their proposed production pathway resolves the global model bias to some degree.*

Response: We changed the description of "current models underestimate" to "previous models underestimate with only gas-phase mechanism included" and acknowledged that this multiphase pathway involving methanediol resolves the global model significantly when we are discussing the significance of multiphase reactions in HCOOH production.

Revision in the main text:
Introduction Section:
Line 60-64: However, with these mechanisms included, HCOOH concentrations remain significantly underestimated by previous models (Baboukas et al., 2000; Bannan et al., 2017; Chaliyakunnel et al., 2016; Le Breton et al., 2012; Millet et al., 2015; Yuan et al., 2015), indicating a substantial missing source of HCOOH.
Line 69-73: A recent chamber experiment has revealed that formaldehyde can be efficiently converted to HCOOH through a multiphase pathway that involves its hydrated form, methanediol. This pathway has been shown to generate up to four times

more formic acid compared to all other known chemical sources combined in a chemistry-climate model, and the modified model largely reproduced observed ambient concentrations of HCOOH (Franco et al., 2021).

*Line 163: Is there any particular reason why a higher endpoint height of 100m was selected for your HYSPLIT simulations?*

Response: The altitude of this site is around 60m. We just used a round number of 100m previously. We re-run the HYSPLIT analysis using altitude of 60m and found the trajectories are almost the same with that of 100m. Nonetheless, we changed it to 60m.

Revision in the main text:
Methods Section 2.2:
Line 151-153: The input parameters were Global Data Assimilation System 1° for the meteorology data; 22.21°N and 114.25°E for the location; and 60 m for the endpoint height.

*Line 176-179: Could you please justify why you set all these parameters to 3? Why are you ignoring gas-phase species in this context? What uncertainty in aerosol properties do you expect from this assumption?*

Response: The input of the ion concentrations was derived from measurement data of Monitor for AeRosols and Gases in ambient Air (MARGA). The ion concentrations (e.g., $Cl^-$ and $NO_3^-$) measured by MAGRA was real ambient value in aerosol phase which had reached equilibrium with gas-phase species in the ambient. In this case, we should set these parameters to 3 to prevent producing corresponding gas-phase species (e.g., HCl and $HNO_3$) and set u to 0 to allow the program looking for all possible solids that can form and include them in the equilibration. The uncertainty of this setting depends on the accuracy of field measurements of aerosol ionic components and to what extent the gas-phase components reach equilibrium with their particle-phase counterparts (e.g., the HCl- $Cl^-$ pair).

*Line 179-180: This statement might be confusing. Please rephrase!*

Revision in the main text:
Methods Section 2.2:
Line 168-169: The aqueous-phase $NO_3^-$ concentration was calculated by dividing the moles of aqueous $NO_3^-$ by the AWC.

*Line 191: What characterizes a typical day? Which typical day criteria did you use? In Figure S3, you provide backward trajectories for many days. Could you please add a separate HYSPLIT analysis to the supplement for the 28 September 2021. It would be useful to color code the backward trajectories by time to understand transport patterns and the influence of different air masses during the day.*

Response: As indicated in the ealier response, in the revised manuscript, we no longer focus on only one day but two-week period for budget analysis. The back trajectory for 28 Sept is shown below for referee's information.

[Figure]

*Line 235-236: Please justify this assumption and discuss related uncertainties. In the result section, you discuss this but the reader is unaware of these details up to this point.*

Response: We agree with this suggestion and added justification of this assumption and related uncertainties in the method section.

Revision in the main text:
Methods Section 2.3:
Line 198-204: We assumed that $\cdot OH$ produced by $NO_3^-$ photolysis was the rate-limiting species due to its significantly lower abundance relative to organics and HCHO was taken as a representative example of potential precursors of HCOOH. However, it is important to note that the production rate of HCOOH from the oxidation of organics may be overestimated in the solution experiment because $\cdot OH$ generated by nitrate photolysis, can also react with other oxidizable species in the ambient atmosphere.

*Line 301: I am confused by the registered trademark sign listed after the Pearson correlation coefficient. What is the meaning of this?*

Response: Sorry for the confusion caused. We changed it to 'r'.

Revision in the main text:
Results Section 3.1:
Line 301-303: Table 2 presents the Pearson correlation coefficients (r) between the concentration of HCOOH and those of other air pollutants or other meteorological parameters during the three distinct periods.

*Line 408: I disagree. The model is not capable of reproducing the steady increase in HCOOH early in the day (from midnight to 9am) nor the rapid decrease in HCOOH*

*after sunset. In addition, I would not label the observed increase from midnight to 2pm as rapid but rather a steady increase.*

Response: We agree and changed the description.
Revision in the main text:
Results Section 3.3:
Line 440-443: Although the modelled concentration with $V_d = 0.50$ cm s$^{-1}$ were similar to observed values, the model failed to accurately predict the timing of the initial increase, peak, and sharp decrease in HCOOH concentrations, indicating that other physical process, such as vertical mixing, also influence the HCOOH concentration.

*Technical corrections:*

*Caption of Table 2: Again, confused by the registered trademark sign.*

Response: We have also corrected it to 'r'.
Revision in Caption of Table 2: Pearson correlation coefficient (r) matrix between the concentration of formic acid and other air pollutants, and related meteorological parameters, during three distinct periods.

*Reference list: The reference order is not according to Copernicus guidelines.*

Response: Previously, we used APA 7$^{th}$ style. Now, we updated the reference list according to Copernicus guidelines which require the published year to be placed at the end of the reference.

Additional revisions:
1. Since we have conducted more sensitivity analysis on modelling, we prefer to discuss all modelling results in "Box model simulation" part and reorganize the manuscript structure as the order of field campaign to laboratory experiments to modelling work. Therefore, we moved "Box model simulation" part to Section 3.3.

2. Due to the modifications made in the methodology and results sections, corresponding revisions have also been made in the abstract and introduction sections as following to ensure consistency in the content:
Abstract:
Line 14-17: Formic acid (HCOOH) is one of the most abundant organic acids in the atmosphere and affects atmospheric acidity and aqueous chemistry. However, the HCOOH sources are not well understood. In a recent field study, we measured atmospheric HCOOH concentrations at a coastal site in South China.
Line 24-26: We incorporated this particle-phase source into a photochemical model and the net HCOOH production rate increased by about three times compared with the default Master Chemical Mechanism (MCM).
Introduction:
Line 60-66: However, with these mechanisms included, HCOOH concentrations

remain significantly underestimated by previous models (Baboukas et al., 2000; Bannan et al., 2017; Chaliyakunnel et al., 2016; Le Breton et al., 2012; Millet et al., 2015; Yuan et al., 2015), indicating a substantial missing source of HCOOH.

In addition to gas-phase production pathways, HCOOH can also be generated through heterogeneous or condensed-phase processes.

Line 82-91: In the condensed organic phase, HCOOH can be produced through the photodegradation of SOA (Henry and Donahue, 2012; Malecha and Nizkorodov, 2016). Additionally, the oxidants such as ·OH, nitrogen dioxide ($NO_2$), and nitrite ions/nitrous acid (HONO) produced from the photolysis of particulate nitrate ($NO_3^-$) can also efficiently oxidise organics to produce HCOOH (Zhang et al., 2021). Apart from laboratory experiments, Paulot et al. (2011) observed a marked positive correlation between HCOOH concentrations and submicron organic aerosol masses in field measurements conducted in three distinct areas: coastal, urban, and polar, and suggested that aerosol aging produces HCOOH. The aforementioned results show that there is a need for a quantitative assessment of the contribution of the photochemical aging of aerosols to HCOOH production in the ambient atmosphere.

3. We updated the DOI link of data shared.
Line 485-486: The data that support the findings of this study are openly available in Zenodo at https://10.5281/zenodo.8415792.

**References mentioned in author's response**:

Guenther, A., Karl, T., Harley, P., Wiedinmyer, C., Palmer, P. I., and Geron, C.: Atmospheric Chemistry and Physics Estimates of global terrestrial isoprene emissions using MEGAN (Model of Emissions of Gases and Aerosols from Nature), Atmos. Chem. Phys, 6, 3181–3210, 2006.

Guenther, A. B., Jiang, X., Heald, C. L., Sakulyanontvittaya, T., Duhl, T., Emmons, L. K., and Wang, X.: The Model of Emissions of Gases and Aerosols from Nature version 2.1 (MEGAN2.1): an extended and updated framework for modeling biogenic emissions, Geosci. Model Dev, 5, 1471–1492, https://doi.org/10.5194/gmd-5-1471-2012, 2012.